# RAMAC: Multimodal Risk-Aware Offline Reinforcement Learning and the Role of Behavior Regularization

**Kai Fukazawa** [1]  **Kunal Mundada** [1]  **Iman Soltani** [1]

## Abstract

In safety-critical domains where online data collection is infeasible, offline reinforcement learning (RL) is attractive only if policies achieve high returns without catastrophic lower-tail risk. Prior work on risk-averse offline RL achieves safety at the cost of either (i) value/model-based pessimism or (ii) restricted policy classes that limit expressiveness, whereas diffusion/flow-based expressive generative policies have largely been used in risk-neutral settings. We introduce **Risk-Aware Multimodal Actor-Critic (RAMAC)**, a simple, modular, model-free framework that couples an expressive generative actor (e.g., diffusion/flow) with a distributional critic and optimizes a composite objective that combines Conditional Value-at-Risk (CVaR) with behavioral cloning (BC), enabling risk-sensitive learning in complex multimodal scenarios. Since out-of-distribution (OOD) actions are a major driver of catastrophic failures in offline RL, we further provide an objective-level analysis showing that controlling behavior divergence via BC suppresses OOD actions and stabilizes CVaR. Instantiating RAMAC with a diffusion actor, we illustrate these insights on a 2-D risky bandit and evaluate on Stochastic-D4RL, observing consistent gains in $\text{CVaR}_{0.1}$ while maintaining strong returns. The code and experimental results are available on the project website.

## 1. Introduction

In high-stakes applications such as autonomous driving, robotics, finance, and healthcare, where real-world exploration can lead to catastrophic consequences, offline RL offers a safe approach for generating policies that not only

[1]University of California, Davis. Correspondence to: Kai Fukazawa <kfukazawa@ucdavis.edu>.

*Proceedings of the 43rd International Conference on Machine Learning*, Seoul, South Korea. PMLR 306, 2026. Copyright 2026 by the author(s).

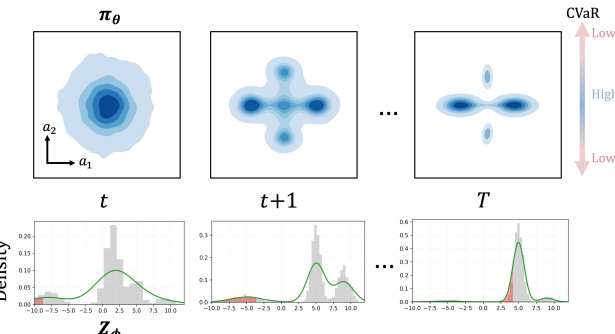

*Figure 1.* **RAMAC learning dynamics (conceptual).** *Top:* The generative policy $\pi_\theta(a \mid s)$ evolves during training; the desired outcome is to retain multiple dataset-supported high-CVaR modes while suppressing low-CVaR modes, without collapsing or moving off support. *Bottom:* The critic $Z_\phi(s, a, \tau)$ models the return distribution, whose lower tail, highlighted in red, defines CVaR. CVaR guides mode reweighting, while BC preserves data support.

maximize long-horizon returns but also *tightly control risk* (Levine et al., 2020). Recent expressive generative policies (Koirala & Fleming, 2026; Park et al., 2025; Wang et al., 2023) can capture multimodal behavior and thus excel in achieving high expected return, yet their primary use has been limited to *risk-neutral* settings. Conversely, existing risk-averse algorithms ensure safety by enforcing conservatism or restricted policy classes, which constrain policy expressiveness (Kumar et al., 2020; Ma et al., 2021; Urpí et al., 2021). This paper asks: *Can we obtain safety without sacrificing expressiveness?*

We answer in the affirmative by proposing the **Risk-Aware Multimodal Actor-Critic (RAMAC)**, a simple and modular framework that couples an expressive generative actor with a distributional critic and optimizes a *single composite objective* that combines BC regularization with distributional risk (instantiated with CVaR). As illustrated conceptually in Fig. 1, the CVaR term steers probability mass away from low-tail regions while BC keeps the generative actor anchored to dataset modes, enabling tail-risk control without sacrificing multimodal expressiveness. This unifies high expressiveness with robust tail-risk control while directly constraining the generative policy to the data support, addressing two central safety concerns in offline RL:

*catastrophic tail outcomes* and *out-of-distribution (OOD) actions*.

Prior offline-RL approaches can be organized by mechanism:

**(1) Policy regularization** constrains the policy to the data manifold via divergence minimization or policy priors, improving stability but often sacrificing policy expressiveness on complex tasks with risk-neutral examples such as (Fujimoto & Gu, 2021; Fujimoto et al., 2019; Kumar et al., 2019; Wu et al., 2019) and risk-aware methods with *prior-anchored perturbation* designs such as (Chen et al., 2025; Urpí et al., 2021).

**(2) Value conservatism** reduces optimistic extrapolation, but can underestimate the value of infrequent yet high-return in-distribution modes due to global pessimism and data imbalance in both risk-neutral (Kumar et al., 2020) and risk-aware instances (Ma et al., 2021).

**(3) Model-based pessimism** bounds transition uncertainty with ensembles and penalties, at the cost of compounding model errors at scale again under both risk-neutral (Rigter et al., 2022; Yu et al., 2020; 2021) and risk-aware (Rigter et al., 2023) settings.

**(4) Expressive generative policies** faithfully clone multimodal behavior and achieve state-of-the-art mean returns, but have so far been used primarily in *risk-neutral* applications (Hansen-Estruch et al., 2023; Kang et al., 2023; Koirala & Fleming, 2026; Park et al., 2025; Wang et al., 2023) including closely related concurrent works that pair diffusion with distributional critics (Liu et al., 2025; Zhang et al., 2025a).

Despite compelling results from these expressive generative policies in risk-neutral RL, their potential in offline risk-aware RL remains largely untapped. Among behavior-regularized risk-averse methods, the most prominent approaches that leverage expressive priors rely on *prior-anchored perturbation* (Chen et al., 2025; Urpí et al., 2021), which trade away much of the multimodal capacity and, as our analysis shows, can still incur OOD actions.

Here, we aim to leverage the advantages of expressive policies with tail-risk control without adding extra learned components (e.g., dynamics models or ensembles) or separate OOD pipelines (e.g., additional risk-averse policy training). To this end, inspired by the success of recent risk-neutral expressive policies such as (Wang et al., 2023; Park et al., 2025), RAMAC couples an expressive generative actor with a distributional critic and *differentiates a single composite objective (BC + CVaR) through the generative process* (Chow et al., 2015; Di Castro et al., 2012). Beyond providing a simple and modular recipe for risk-aware expressive policies, we give an objective-level analysis that links

behavior divergence control to safety: BC regularization suppresses per-state OOD action probabilities and, more importantly, stabilizes CVaR of bounded return-related scores, explaining why this *single composite objective* mitigates lower-tail blow-ups for expressive generative actors (Sec. 4). We also empirically show that RAMAC yields high expected return while minimizing risk on complex multimodal offline benchmarks. Our contributions can be summarized as:

1) Risk-aware expressive policy learning: We introduce **RA-MAC**, a simple, modular, and model-free framework that enables risk-aware learning with expressive generative actors by optimizing a *single composite objective* that combines BC regularization with distributional risk optimization. Our primary instantiation is a diffusion-based actor, **RADAC (Risk-Aware Diffusion Actor-Critic)**.

2) Objective-level and geometric analysis: We give simple bounds linking BC-induced behavior divergence control to (i) per-state OOD action events and (ii) the stability of $\text{CVaR}_\alpha$ for bounded utility scores. We also provide a geometric perspective showing that prior-anchored perturbation can still allocate probability mass off-support even with expressive priors.

3) Experimental evaluation: On Stochastic-D4RL benchmarks, **RADAC** outperforms baselines on CVaR while maintaining competitive mean return on most tasks. We additionally report a flow-matching variant, **RAFMAC (Risk-Aware Flow-Matching Actor-Critic)**, in App. E.2, which shows similar trends. Finally, we (i) visualize the geometric OOD mechanism in a 2-D risky bandit that contrasts *prior-anchored perturbation* and *expressive generative policies*, and (ii) use OOD-action detectors on Stochastic-D4RL to empirically validate the analysis.

## 2. Preliminaries

**Offline RL.** We consider a finite-horizon Markov Decision Process (MDP) $\mathcal{M} = (\mathcal{S}, \mathcal{A}, P, r, \gamma, H)$ with state space $\mathcal{S}$, action space $\mathcal{A}$, transition kernel $P(\cdot \mid s, a)$, reward function $r(s, a)$, discount factor $\gamma \in (0, 1)$, and horizon $H \in \mathbb{N}$ (Sutton et al., 1998). In offline RL, the learner is given only a static dataset $\mathcal{D} = \{(s_i, a_i, r_i, s_i')\}_{i=1}^N$ collected by some unknown behavior policy $\beta$, and cannot further interact with the environment (Prudencio et al., 2023). Let $\text{supp}(\mathcal{D})$ denote the empirical support of the dataset. The objective is to learn a policy $\pi$ that maximizes the expected return $J(\pi) = \mathbb{E}_{\pi, P}[\sum_{t=0}^{H-1} \gamma^t r_t]$ without extra environment interaction. The central challenge is *distributional shift* (OOD): When $\pi$ visits $(s, a) \notin \text{supp}(\mathcal{D})$, value estimates $Q(s, a)$ extrapolate and can become arbitrarily inaccurate (Kumar et al., 2020). Policies that place non-negligible mass on such OOD actions may therefore suffer catastrophic failures at deployment. Prior work alleviates

this issue with behavior regularization, conservative critics, or model-based pessimism.

**Behavior–Regularized Actor–Critic (BRAC).** A large family of offline methods uses an actor–critic with an explicit proximity term to the behavior policy (Fujimoto & Gu, 2021; Kumar et al., 2019; Nair et al., 2020; Wu et al., 2019). A representative actor–critic objective takes the form:

$$\mathcal{L}_{\text{Actor}}(\theta) = \mathbb{E}_{\substack{(s,a)\sim\mathcal{D}, \\ a^\pi\sim\pi_\theta(\cdot|s)}} \big[ -Q_\phi(s, a^\pi) - \alpha \log \pi_\theta(a \mid s) \big],$$
(1)

$$\mathcal{L}_{\text{Critic}}(\phi) = \mathbb{E}_{\substack{(s,a,r,s')\sim\mathcal{D}, \\ a'\sim\pi_\theta(\cdot|s')}} \big[ Q_\phi(s, a) - r - \gamma Q_{\bar\phi}(s', a') \big]^2.$$
(2)

Here the second term $-\alpha \log \pi_\theta(a \mid s)$, evaluated on dataset actions $a \sim \mathcal{D}$, plays the role of a *behavior regularizer*: it is typically instantiated as a behavioral-cloning (BC) term or another proximity/divergence penalty that keeps $\pi_\theta(\cdot \mid s)$ close to the behavior policy $\beta(\cdot \mid s)$. [1] Empirically, BRAC-style objectives have turned out to be surprisingly strong in offline RL (Tarasov et al., 2023). In this work, we extend this behavior-regularized pattern to a *distributional* actor–critic in which the critic is expanded into a *distributional critic* that models the return distribution, enabling optimization of tail-sensitive objectives such as CVaR in place of the mean $Q$.

**Distributional RL and Risk Measures.** Standard actor–critic methods including BRAC optimize the expected return by learning the mean action-value function $Q^\pi(s, a) = \mathbb{E}[Z^\pi(s, a)]$ as shown in Eq. 2. Distributional RL instead models the entire *return distribution* $Z^\pi(s, a)$ (Bellemare et al., 2017). The distributional Bellman operator is:

$$(\mathcal{T}^\pi Z)(s, a) \overset{d}{=} r(s, a) + \gamma Z(s', a').$$
(3)

where $s' \sim P(\cdot|s, a)$ and $a' \sim \pi(\cdot|s')$. A common parameterization uses an Implicit Quantile Network (IQN) (Dabney et al., 2018) to approximate the inverse cumulative distribution function (CDF) $Z_\phi(s, a; \tau) \approx F^{-1}_{Z^\pi(s,a)}(\tau)$ for quantile levels $\tau \in (0, 1)$. Access to quantiles enables distortion-based risk objectives that emphasize different parts of the return distribution. Risk sensitivity in RL can be formulated in multiple ways; in this work, we focus specifically on Conditional Value-at-Risk (CVaR) as our primary instantiation because it directly targets catastrophic lower-tail outcomes.

---

[1] Following Park et al. (2025), we use the term *behavior-regularized actor–critic* broadly for offline actor–critic methods that combine value improvement with behavior regularization, rather than only for the original BRAC algorithm. For expressive generative actors, the behavior-fidelity term is implemented using the actor's native BC surrogate: denoising loss for RADAC and flow-matching loss for RAFMAC.

For a risk level $\alpha \in (0, 1]$, the CVaR admits the integral form used for actor gradients:

$$\text{CVaR}_\alpha(X) = \frac{1}{\alpha} \int_0^\alpha F^{-1}_X(\tau) \, d\tau.$$
(4)

In our analysis, we will also later consider CVaR of bounded scalar cost/risk scores, for which simple stability bounds are available. Optimizing $\text{CVaR}_\alpha$ encourages policies that trade some mean performance for improved behavior in the worst $\alpha$-fraction of trajectories, which is crucial in safety-critical settings. In RAMAC, the distributional critic provides quantile estimates from which CVaR and its gradients with respect to actions can be computed and backpropagated through the policy.

**Expressive Generative Policies as Differentiable Trajectories.** Recent offline RL methods stay within the behavior-regularized actor-critic template of Eqs. 1 and 2, but replace the simple parametric actor with an expressive conditional generative model (Hansen-Estruch et al., 2023; Kang et al., 2023; Koirala & Fleming, 2026; Park et al., 2025; Wang et al., 2023). Given a state $s$ and latent $z \sim \mathcal{N}(0, I)$, the policy generates an action $a = \psi_\theta(s, z)$ along a *differentiable path* (Hansen-Estruch et al., 2023; Wang et al., 2023), while an explicit behavior term keeps $\psi_\theta$ close to the dataset actions. We focus on the two families:

(i) *Diffusion policies* model a reverse-time stochastic differential equation (SDE) over actions (Song et al., 2021),

$$d\mathbf{a}_t = f_\theta(t, \mathbf{a}_t, s) \, dt + g(t) \, d\mathbf{w}_t,$$
(5)

where a forward noising process gradually corrupts dataset actions into near-Gaussian noise, and the network $f_\theta$ learns to reverse this process conditioned on the state $s$.

(ii) *Flow-matching policies* solve a deterministic ODE (Lipman et al., 2022),

$$\frac{d\mathbf{a}_t}{dt} = v_\theta(t, \mathbf{a}_t, s).$$
(6)

where a neural vector field $v_\theta$ transports samples from a simple base distribution to the data distribution along a continuous trajectory. Integrating Eq. 6 from an initial noise sample yields an action conditioned on $s$.

In both cases, the overall map $\psi_\theta : (s, z) \mapsto a$ is fully differentiable, and the behavior term encourages $\psi_\theta$ to approximate the behavior action distribution itself; the critic then fine-tunes this expressive behavior model using scalar signals. Prior work typically uses expected-value or advantage-based signals from a mean-value critic to update the generative policy under a BRAC template (as in Eq. 1), yielding a *risk-neutral* generative actor-critic method. In contrast, RAMAC replaces the mean critic with a *distributional* critic and

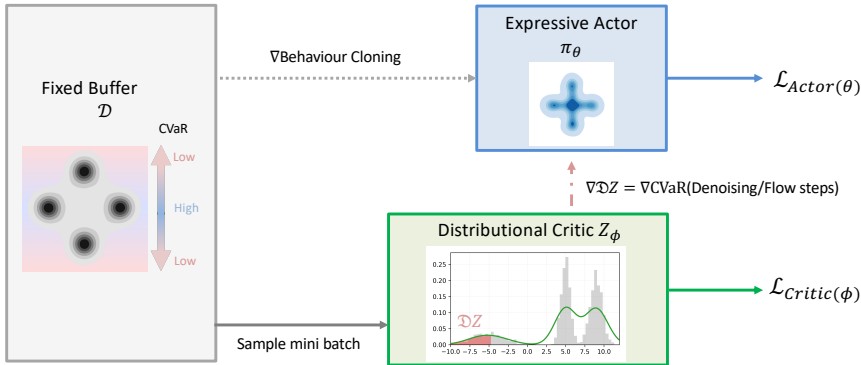

*Figure 2.* **RAMAC pipeline.** From the offline buffer $\mathcal{D}$ (gray), the distributional critic $Z_\phi$ (green) fits the return law with a quantile loss and aggregates its lower tail into a CVaR signal. That signal is differentiated through the generative path of the actor $\pi_\theta$ (blue; diffusion or flow), which is trained with the composite objective $\mathcal{L}_\pi = \mathcal{L}_{\text{BC}} + \eta\,\mathcal{L}_{\text{Risk}}$ to shift mass away from low-quantile regions while staying on-manifold, yielding stable lower-tail performance.

uses tail-sensitive risk signals such as $\text{CVaR}_\alpha(Z^\pi(s,a))$ (Eq. 4) to shape the same generative policy under a *single composite objective* that combines BC regularization and distributional risk; the exact loss is introduced in Sec. 3.

## 3. Method

We now introduce the **Risk-Aware Multimodal Actor-Critic (RAMAC)**. RAMAC is a simple, modular, model-free framework that couples (i) a *distributional critic* learning the conditional return law and (ii) an *expressive generative actor* (diffusion or flow) trained by a *single composite objective*. As summarized in Fig. 2, the critic provides a CVaR signal, whose gradients are backpropagated through the actor's differentiable generative path, while a BC term regularizes the policy to the dataset support. Intuitively, CVaR updates steer probability mass away from low-quantile, catastrophic regions, whereas BC preserves multimodal high-reward behavior by constraining the behavior divergence. This BRAC-style regularizer not only mitigates OOD actions but also stabilizes tail objectives; our objective-level bounds in Sec. 4 formalize these effects.

### 3.1. Distributional Critic

Risk-sensitive objectives such as CVaR require access to the entire return distribution. We therefore adopt a distributional critic $Z_\phi$ via IQN (Dabney et al., 2018), building on the distributional Bellman operator in Eq. 3. Here $\delta_\phi$ is the *quantile TD error* that matches the predicted quantile $Z_\phi(s,a;\tau)$ to the distributional Bellman target $r + \gamma Z_{\bar\phi}(s',a';\tau')$ with independently sampled quantile levels $\tau, \tau'$. We minimize a distributional Bellman residual:

$$\delta_\phi := r + \gamma Z_{\bar\phi}(s',a';\tau') - Z_\phi(s,a;\tau),$$

$$\mathcal{L}_{\text{Critic}}(\phi) = \mathbb{E}_{\substack{(s,a,r,s')\sim\mathcal{D}, \\ a'\sim\pi_\theta(\cdot|s'),\,\tau,\tau'\sim\mathcal{U}(0,1)}}\Big[\mathcal{L}_\kappa\big(\delta_\phi\,;\,\tau\big)\Big]. \quad (7)$$

where $\mathcal{L}_\kappa(\delta;\tau)$ denotes the IQN quantile-Huber loss (with $\kappa = 1$) (Dabney et al., 2018); we provide its explicit form and implementation details in App. D. This yields lower-tail quantiles that will directly drive the risk-aware actor update in Sec. 3.2.

### 3.2. Risk-Aware Generative Actor

Given a state $s$ and latent noise $z \sim \mathcal{N}(0, I)$, the generative actor produces an action $a = \psi_\theta(s, z)$. We define CVaR at level $\alpha$ through the critic's quantiles and use a Monte Carlo estimator:

$$\begin{aligned}\text{CVaR}_\alpha\big(Z_\phi(s,a)\big) &= \frac{1}{\alpha}\int_0^\alpha Z_\phi(s,a;\tau)\,d\tau \\ &\approx \frac{1}{K}\sum_{k=1}^K Z_\phi\big(s,a;\tau_k\big),\end{aligned} \quad (8)$$

where $\tau_k \sim \mathcal{U}(0,\alpha)$ for $k = 1,\ldots,K$. The risk loss maximizes this quantity. This is equivalent to minimizing the negative CVaR [2]:

$$\mathcal{L}_{\text{Risk}}(\theta) = -\mathbb{E}_{s\sim\mathcal{D},\,a\sim\pi_\theta(\cdot|s)}\big[\text{CVaR}_\alpha\big(Z_\phi(s,a)\big)\big]. \quad (9)$$

### 3.3. Behavior-Regularized Objective

The complete policy objective balances risk aversion with fidelity to the offline dataset. We instantiate fidelity through a standard behavior cloning (BC) term that encourages the policy to reproduce the behavior distribution. We define:

$$\mathcal{L}_{\text{BC}}(\theta) = -\mathbb{E}_{(s,a)\sim\mathcal{D}}\big[\log\pi_\theta(a\mid s)\big], \quad (10)$$

which corresponds to the BRAC-style behavior regularizer in Eq. 1 up to a scaling of the coefficient. It combines

---

[2]This specific loss, instantiated with CVaR, is what we refer to as $\mathcal{L}_{\text{CVaR}}$ in our architectural diagrams for clarity.

**Algorithm 1 Risk-Aware Multimodal Actor-Critic (RA-MAC)**

**Initialize** policy network $\pi_\theta$, critic $Z_\phi$, target critic $Z_{\bar{\phi}}$; mini-batch size $B$, risk level $\alpha$, critic-tail samples $K$, Exponential Moving Average (EMA) rate $\rho$.

**repeat**

    Sample a mini-batch $\{(s, a, r, s')\}_{b=1}^B \sim \mathcal{D}$.

    **Training Critic:**

    Sample $z' \sim \mathcal{N}(0, I)$ and set $a' = \psi_\theta(s', z')$;

    Sample $\tau, \tau' \sim \mathcal{U}(0, 1)$

    Update $\phi$ by minimizing $\mathcal{L}_{\text{Critic}}(\phi)$ *(Eq. 7)*.

    **Training Actor:**

    Sample $z \sim \mathcal{N}(0, I)$ and set $a = \psi_\theta(s, z)$;

    Sample $\tau_1, \ldots, \tau_K \sim \mathcal{U}(0, \alpha)$

    Update $\theta$ by minimizing $\mathcal{L}_\pi(\theta)$ *(Eq. 11)*.

    **Target update:** $\bar{\phi} \leftarrow \rho\,\bar{\phi} + (1 - \rho)\,\phi$.

**until** converged

the risk term with a standard behavior cloning (BC) loss, $\mathcal{L}_{\text{BC}}(\theta)$:

$$\mathcal{L}_\pi(\theta) = \underbrace{\mathcal{L}_{\text{BC}}(\theta)}_{\text{data fidelity}} + \eta \underbrace{\mathcal{L}_{\text{Risk}}(\theta)}_{\text{tail-risk aversion}}. \qquad (11)$$

where $\eta$ is a hyperparameter. Our primary instantiation is a diffusion policy (**RADAC**), while an additional flow-matching variant (**RAFMAC**) is reported in App. E.2. We show pseudocode for RAMAC in Algorithm 1 and describe the full implementation details in App. D

## 4. Behavior Regularization in Offline RL

Prior work has demonstrated the importance of behavior regularization in offline RL due to its ability to constrain the learned policy to the data manifold and curb value extrapolation (Tarasov et al., 2023). A commonly adopted regularization scheme in offline risk-aware RL is the prior-anchored perturbation method (e.g., ORAAC, UDAC)[3] (Chen et al., 2025; Urpí et al., 2021), which uses a linear mixing of actions from a pretrained BC policy and from the RL actor (a bounded residual). Here, we first discuss a limitation of this regularization approach. We then highlight the advantages of our scheme, namely, the behavior-regularized objective in Eq. 11.

### 4.1. Prior-Anchored Perturbation and Its Limitations

In this approach, policy output can be written as:

$$a = b + \zeta_\psi(s, b), \quad \|\zeta_\psi(s, b)\| \le \Phi, \qquad (12)$$

---

[3]For simplicity and consistency with our experiments, we will refer to UDAC as *ORAAC–Diffusion*

where $b \sim G_\phi(\cdot \mid s)$, $\zeta_\psi$ is a *learned residual* (optimized to increase $Q$ or CVaR) and the norm bound $\Phi$ *keeps updates close to the anchor*. Define the anchor support $\mathcal{S}_G(s)$ (the region in action space where $G_\phi(\cdot \mid s)$ places mass), the full action space $\mathbb{R}^d$, and the $\Phi$-radius ball of $b$

$$B_\Phi(b) = \{a \in \mathbb{R}^d : \|a - b\|_2 \le \Phi\}. \qquad (13)$$

where any perturbed action $a = b + \zeta_\psi$ with $\|\zeta_\psi\| \le \Phi$ lies in $B_\Phi(b)$. Hence on-manifold deployment is guaranteed by the *safety margin* condition

$$\text{dist}(b, \mathbb{R}^d \backslash \mathcal{S}_G(s)) > \Phi \implies \begin{cases} B_\Phi(b) \subseteq \mathcal{S}_G(s), \\ \forall \|\zeta_\psi\| \le \Phi : a \in \mathcal{S}_G(s). \end{cases} \qquad (14)$$

where $\text{dist}(x, A) := \inf_{y \in A} \|x - y\|_2$ denotes Euclidean distance. OOD can still occur when this margin fails. This method provides a convenient *local* improvement rule; however, prior work has observed that it suffers from *poor mode coverage* in multimodal action spaces (Wang et al., 2023). In addition to the identified limitations, we show *a distinct geometric weakness* that can occur even without multimodality; having multiple modes merely magnifies the effect.

**Lemma 4.1.** *Fix $s$ and write $I_s = \mathcal{S}_G(s)$ and $O_s = \mathbb{R}^d \setminus I_s$. Suppose there exist an anchor $b^\star \in I_s$, a radius $\Phi > 0$, and a measurable leakage region $A_s \subseteq B_\Phi(b^\star) \cap O_s$ such that $\lambda(A_s) > 0$. If the policy $\pi_{\text{anch}}(\cdot \mid s)$ induced by Eq. 12 admits a density $p(\cdot \mid s)$ satisfying*

$$p(a \mid s) \ge c > 0 \quad \text{for all } a \in A_s,$$

*then its per-state OOD probability*

$$\delta_s(\pi_{\text{anch}}) := \pi_{\text{anch}}(O_s \mid s)$$

*satisfies*

$$\delta_s(\pi_{\text{anch}}) \ge c \cdot \lambda(A_s) > 0.$$

*In particular, as long as the induced density remains bounded below by $c > 0$ on $A_s$ during residual training, further optimization of the residual cannot drive $\delta_s(\pi_{\text{anch}})$ to zero. Proof appears in App. B.1.*

Lemma 4.1 is a conditional mechanism statement: whenever an anchor-centered perturbation policy assigns nontrivial probability density to an off-support portion of its feasible perturbation region, residual training alone cannot eliminate the associated OOD probability while that local mass persists. Thin or nonconvex behavior supports can make such overlap more likely, since bounded residual updates are not explicitly constrained to remain on the data manifold (see App. B.1.1 for a more detailed discussion).

### 4.2. Behavior-Regularized Objective: Why It Works Better

In contrast to prior-anchored perturbation, RAMAC applies behavior regularization *directly* to the *deployed* generative

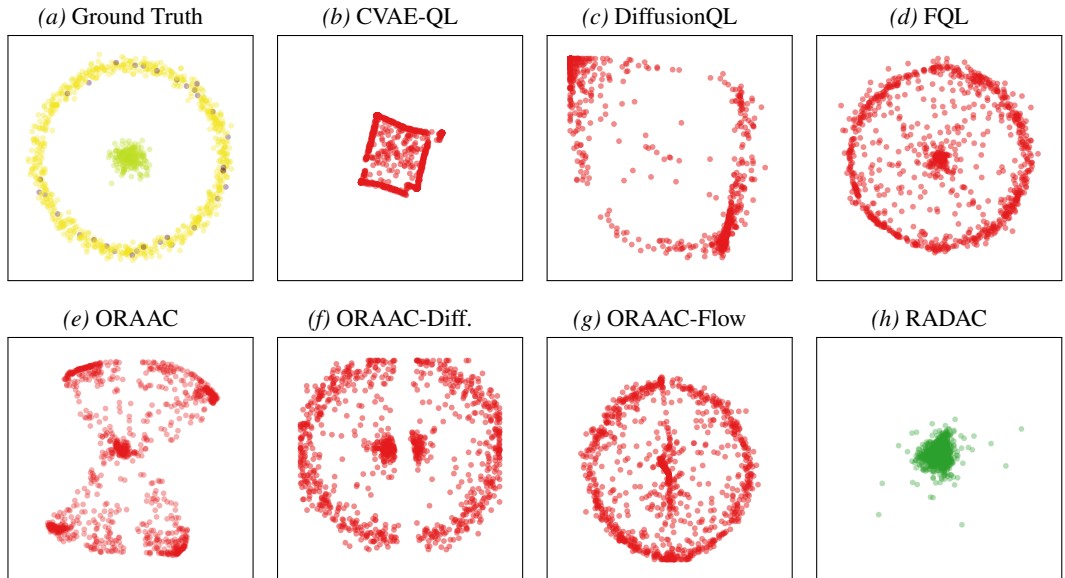

*Figure 3.* **Toy Risky Bandit Results** *Top:* Ground truth consists of a safe center mode yellow-green and a risky ring where high-reward samples yellow are interspersed with catastrophic penalties (purple). Risk-neutral generative baselines concentrate on the risky ring or collapse topology. *Bottom:* Prior-anchored perturbation methods produce samples in the low-density inter-mode region, exhibiting OOD leakage. RADAC concentrates near the safe center without collapsing into the low-density bridge / without filling the inter-mode gap. See App. E.1 for more results.

policy $\pi_\theta$ (Eq. 11). For explicit-likelihood actors, the BC term corresponds to the negative log-likelihood and thus controls the forward KL $D_{\text{KL}}(\beta(\cdot \mid s)\|\pi_\theta(\cdot \mid s))$ up to an additive constant. For our diffusion and flow-matching instantiations, we instead use their native BC-style surrogates, namely denoising and flow-matching losses, which anchor the generative process to dataset actions (see footnote in Sec. 2). The following KL-based results therefore formalize the idealized divergence-controlled mechanism, whose OOD implication we evaluate empirically in Sec. 5.3.

**Lemma 4.2.** *For each state $s$, let $I_s = \{a : \beta(a \mid s) > 0\}$ and $O_s = I_s^c$. Assume $\beta \ll \pi_\theta$ (absolute continuity on $I_s$). Then the per-state OOD probability $\delta_s(\pi_\theta) := \pi_\theta(O_s \mid s)$ satisfies*

$$\delta_s(\pi_\theta) \leq 1 - \exp\left\{-D_{\text{KL}}\big(\beta(\cdot \mid s) \,\|\, \pi_\theta(\cdot \mid s)\big)\right\}. \quad (15)$$

This formalizes a simple point: shrinking the forward KL suppresses probability mass outside the behavior support. Proof appears in App. B.2. We can extend this risk-agnostic mechanism to bound its impact on tail risk objectives.

**Proposition 4.3.** *Fix a state $s$. Let $X_s : \mathcal{A} \to [m, M]$ be any bounded utility score, such as a clipped return or critic signal (equivalently, the negative of a bounded cost). For $\alpha \in (0, 1]$ and any two action distributions $p(\cdot \mid s)$ and*

$q(\cdot \mid s)$, *letting $A \sim p(\cdot \mid s)$ and $A' \sim q(\cdot \mid s)$,*

$$\begin{aligned}
&\big|\text{CVaR}_\alpha(X_s(A)) - \text{CVaR}_\alpha(X_s(A'))\big| \\
&\leq \frac{M - m}{\alpha} \text{TV}\big(p(\cdot \mid s), q(\cdot \mid s)\big) \\
&\leq \frac{M - m}{\alpha} \sqrt{\tfrac{1}{2} D_{\text{KL}}\big(q(\cdot \mid s) \,\|\, p(\cdot \mid s)\big)}. \quad (16)
\end{aligned}$$

**Corollary 4.4.** *Under the assumptions of Prop. 4.3, taking $q = \beta(\cdot \mid s)$ and $p = \pi_\theta(\cdot \mid s)$ yields*

$$\begin{aligned}
&\big|\text{CVaR}_\alpha^\beta - \text{CVaR}_\alpha^{\pi_\theta}\big| \\
&\leq \frac{M - m}{\alpha} \sqrt{\tfrac{1}{2} D_{\text{KL}}\big(\beta(\cdot \mid s) \,\|\, \pi_\theta(\cdot \mid s)\big)}. \quad (17)
\end{aligned}$$

Proposition 4.3 and Corollary 4.4 highlight that CVaR is amplified by $1/\alpha$ under distribution shift. This makes even small behavior mismatch potentially catastrophic in the lower tail, motivating explicit behavior regularization when optimizing $\text{CVaR}_\alpha$. Proof is in App. B.3.

### 4.3. Toy Risky Bandit Example

We design a 2-D contextual bandit with two disjoint modes (*Toy Risky Bandit*) to make the above geometric analysis concrete: The top-left panel in Fig. 3 shows a ground truth that consists of a *safe center* (moderate reward, no catastrophic tail) and a *risky ring* (higher mean with rare large penalties). The task isolates multimodality and lower-tail hazards. Below we introduce our baselines.

*Table 1.* Stochastic–D4RL results over 5 seeds. We report Mean and CVaR$_{0.1}$; best in dark/ second in light shaded. Additional fixed-checkpoint results are provided in App. E.2.

| Dataset | Metric | CQL | CODAC | ORAAC | ORAAC-Diff. | FQL | DiffusionQL | RADAC |
|---|---|---|---|---|---|---|---|---|
| HalfCheetah-M-E | Mean | −66.66 | −0.12 | 796.06 | 650.70 | 844.14 | −20.71 | **916.64** |
| | CVaR | −135.39 | −0.11 | 742.94 | 455.40 | 754.44 | −76.39 | **805.25** |
| Walker2d-M-E | Mean | −21.52 | 23.96 | 969.62 | 823.20 | 1309.48 | −32.38 | **1708.68** |
| | CVaR | −64.88 | −43.88 | 358.55 | 317.92 | 468.15 | −116.19 | **573.22** |
| Hopper-M-E | Mean | −25.87 | 26.59 | 714.15 | 577.73 | 341.16 | −279.97 | **1277.74** |
| | CVaR | −111.37 | −150.92 | 374.63 | 240.48 | −8.80 | −872.95 | **800.64** |
| HalfCheetah-M-R | Mean | −66.21 | −0.11 | 18.99 | 220.00 | 434.33 | 279.95 | **525.84** |
| | CVaR | −127.09 | −1.47 | −34.09 | 44.93 | 224.73 | 79.93 | **278.65** |
| Walker2d-M-R | Mean | −16.90 | 33.59 | 126.94 | 4.62 | 411.36 | 96.88 | **615.94** |
| | CVaR | −51.49 | −52.63 | −203.64 | −113.20 | 5.08 | 48.14 | **145.21** |
| Hopper-M-R | Mean | −16.25 | −47.83 | −18.00 | 25.68 | 373.16 | −2.79 | **385.58** |
| | CVaR | −118.70 | −160.08 | −129.25 | −131.91 | −62.24 | −51.33 | **−8.16** |

### 4.3.1. EXPRESSIVE BUT RISK-NEUTRAL CONTROLS

As a common notation, let $G_\phi(\cdot \mid s)$ denote an expressive conditional generator (diffusion model, flow, or conditional VAE) that induces the policy $a \sim G_\phi(\cdot \mid s)$. Our risk-neutral expressive baselines, DiffusionQL (Wang et al., 2023), FlowQL (FQL) (Park et al., 2025), and a conditional VAE-QL (CVAE-QL), all minimize

$$\mathcal{L}_{\mathrm{RN}}(\phi) = \lambda_{\mathrm{BC}} \, \mathbb{E}_{(s,a)\sim\mathcal{D}}\big[\ell\big(a, G_\phi(s)\big)\big] - \mathbb{E}_{\substack{s\sim\mathcal{D},\\ a\sim G_\phi(\cdot|s)}}\big[Q_\psi(s,a)\big]. \quad (18)$$

i.e., a standard BC loss plus a risk-neutral value-improvement term on top of an expressive actor $G_\phi$ with scalar critic $Q_\psi$. Each method simply instantiates $G_\phi$ (diffusion, flow, or CVAE) and its optimization hyperparameters; full details are given in App. F.1.

### 4.3.2. PRIOR-ANCHORED PERTURBATION

As risk-aware anchor–perturbation baselines we use ORAAC, ORAAC–Diffusion (ORAAC-Diff.) (Chen et al., 2025; Urpí et al., 2021), and ORAAC–Flow, all instantiated via Eq. 12 with their original coherent risk objectives. Concretely, given a behavior anchor $b \sim \beta(\cdot \mid s)$, each method learns a residual perturbation $f_\phi$ as in Sec. 4 and optimizes its own risk functional (e.g., CVaR or distorted expectations) under this prior-anchored geometry.

### 4.3.3. EXAMPLE RESULTS

The resulting policy distributions for various methods are shown in Fig. 3.

Risk-neutral expressive controls (Fig. 3 (b-d)): Overall, as expected, these methods are risk-blind and chase high-$Q$ ridges ignoring the lower tail. FQL often preserves both modes but does not suppress mass on the hazardous ring;

DiffusionQL drifts toward sparsely covered high-$Q$ pockets on the ring, yielding risk exposure; the CVAE variant collapses topology and fills low-density bridges.

Prior-anchored perturbation (Fig. 3 (e-g)): ORAAC and its diffusion/flow variants place substantial mass in the inter-mode low-density region, regardless of whether the BC prior is expressive or not. These controlled toy results illustrate the geometric leakage mechanism in Lemma 4.1.

RADAC (Fig. 3 (h)): By sending CVaR signals from a distributional critic through the diffusion trajectory while regularizing with BC, RADAC concentrates probability near the safe center without filling the gap. Full configuration and additional plots are in App. E.1 and F.1. Overall, these patterns empirically support our analysis.

## 5. Experiments

In this section, we evaluate **RADAC** on the Stochastic-D4RL benchmarks to validate both *risk awareness* and *policy expressiveness*. We also quantify the OOD action rate $\varepsilon_{\mathrm{act}}$ (Sec. 5.3) to empirically validate the behavior-divergence mechanism in Sec. 4. Additional results, including the flow-matching instantiation **RAFMAC**, appear in App. E.2.

**Tasks** We augment standard D4RL locomotion tasks (Fu et al., 2020) with rare heavy-tailed penalties tied to forward velocity (HalfCheetah) or torso pitch angles (Hopper, Walker2d), together with early termination when the torso leaves a healthy range, following (Urpí et al., 2021). We evaluate on HOPPER, WALKER2D, and HALFCHEETAH using the MEDIUM–EXPERT (M-E) and MEDIUM–REPLAY (M-R) datasets, which are typically multimodal due to mixed data-collection policies. This lets us examine whether RAMAC learns *risk-aware* policies *without sacrificing mul-*

*timodality*. Full construction details appear in App. F.2.

**Baselines** We compare against representative offline-RL methods covering value conservatism, distributional conservatism, anchor-perturbation risk aversion, and risk-neutral expressive generators: CQL (Kumar et al., 2020), CO-DAC (Ma et al., 2021), ORAAC (Urpí et al., 2021), ORAAC-Diffusion (ORAAC-Diff.) (Chen et al., 2025), DiffusionQL (Wang et al., 2023), and FlowQL (FQL) (Park et al., 2025). We defer detailed configurations to App. F.3.

**Evaluation** Following the protocols adopted in (Urpí et al., 2021; Wang et al., 2023), we train for 2000 epochs, each with 1000 gradient steps. We evaluate methods at fixed intervals of gradient steps and report (i) raw returns averaged over 5 seeds and (ii) episodic $\text{CVaR}_{0.1}$ computed over 50 rollouts (10 evaluation episodes per seed) to avoid score normalization artifacts. For ORAAC and CODAC, we adopt the authors' risk-aware objectives (risk level $\alpha=0.1$). For the other baselines, we tune hyperparameters to ensure fairness and otherwise use authors' recommended settings. Further protocol details appear in App. F.2. Runtime and inference-latency comparisons for RADAC with recent baselines are reported in App. E.4.

### 5.1. Results and Analysis

Table 1 reports Mean and $\text{CVaR}_{0.1}$ for RADAC alongside baselines. Across six tasks, RADAC delivers *strong lower tails with competitive or higher means*. FQL is often the strongest risk-neutral baseline, suggesting that flow-based generators can stabilize training even without explicit tail-risk shaping, but its lower-tail returns remain below RADAC. Further analysis is provided in the appendix, including an ablation on the tail-sample budget $K$ in the CVaR estimator (App. E.3.3), an objective ablation separating the BC and CVaR terms (App. E.3.1), and a comparison of alternative risk distortions (App. E.3.2).

### 5.2. Qualitative Safety Analysis

We further visualize a contrast among three representative methods: risk-aware expressive generator RADAC, risk-neutral expressive generator DiffusionQL, and the anchor-perturb risk-averse method ORAAC. Fig. 4 plots the monitored distribution of policies against safe regions. RADAC concentrates probability mass inside or near the risk-free boundary while *actively reallocating probability to high-return modes that lie within these regions*. DiffusionQL is tightly concentrated around zero because rare, high penalties depress bootstrapped values near the risk-free boundary. On the other hand, ORAAC regularizes toward a behavior anchor and thus can place nontrivial mass in low-density inter-mode regions due to its anchored-perturbation geometry. These features can be further ob-

served in App. E.2 which plots Pareto frontiers of mean return versus per-episode safety-violation counts. Additionally, we show empirical return distributions and risk–return frontiers in App. E.5 and App. E.6.

### 5.3. Empirical Validation of the OOD Mechanism

We now provide measurements of OOD actions to validate the insights in Sec. 4. For each policy, we report $\varepsilon_{\text{act}}$, the fraction of evaluation state-action pairs $(s_t, a_t)$ whose *state-conditioned* 1-NN action distance exceeds $\kappa \times \text{median } d_{\text{NN}}$. Sec. 4 predicts that (i) the BC-regularized CVaR objective should reweight probability *within* the data manifold, yielding low $\varepsilon_{\text{act}}$, and (ii) ORAAC variants, being based on anchor–perturbation, should exhibit *higher* $\varepsilon_{\text{act}}$ than RADAC; Table 2 confirms this prediction: RADAC retains low OOD across tasks, whereas ORAAC variants are consistently higher, even with an expressive diffusion prior, matching the expected geometric leakage from anchor–perturbation. Similar RADAC < ORAAC rankings hold under alternative OOD detectors (App. E.3.4). RADAC achieves risk awareness and expressiveness simultaneously while maintaining low OOD rates.

*Table 2.* State-conditioned OOD action rate (%± std over 3 seeds) on MEDIUM−EXPERT ($\kappa=3$).

| Task | RADAC | ORAAC | ORAAC-Diff. |
|---|---|---|---|
| HalfCheetah | $1.22\pm0.21$ | $29.27\pm12.76$ | $35.4\pm25.9$ |
| Walker2d | $0.96\pm0.45$ | $5.57\pm3.31$ | $18.5\pm2.71$ |
| Hopper | $1.21\pm0.21$ | $11.60\pm3.95$ | $22.7\pm6.56$ |

## 6. Conclusions

This paper introduces RAMAC, a model-free framework for risk-aware offline RL with expressive generative policies. RAMAC couples a distributional critic with a diffusion/flow actor and optimizes a single composite objective that combines behavioral cloning (BC) with CVaR. Our analysis highlights two complementary mechanisms: (i) prior-anchored perturbation can retain off-support leakage on thin or nonconvex supports when its feasible perturbation region overlaps the support complement and the induced policy maintains mass there, and (ii) controlling behavior divergence of the deployed policy suppresses OOD action probability and stabilizes tail objectives such as $\text{CVaR}_\alpha$ under distribution shift. On Stochastic-D4RL benchmarks, RADAC achieves consistently stronger lower-tail returns ($\text{CVaR}_{0.1}$) while maintaining competitive mean performance and lower measured OOD action rates than representative anchor-perturbation risk-aware baselines.

## Impact Statement

This work advances risk-aware learning for decision-making policies, with the goal of improving reliability under distri-

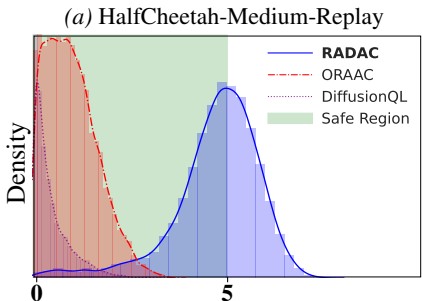 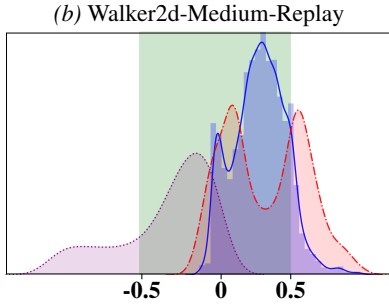 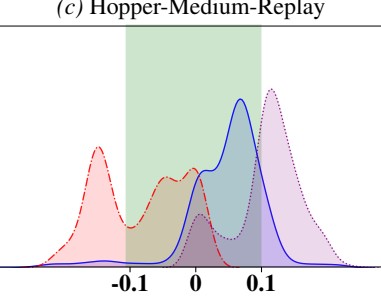

*Figure 4.* Policy distributions for RADAC, ORAAC, and DiffusionQL; shaded bands indicate safe operational ranges (HalfCheetah: $v \leq 5$; Hopper: $|\theta| \leq 0.1$; Walker2d: $|\theta| \leq 0.5$). RADAC reduces mass beyond thresholds.

bution shift and rare adverse outcomes. Potential positive impacts include safer deployment of learned controllers in domains such as robotics and autonomous systems, and more informative evaluation of tail risks.

As with most learning-based decision systems, misuse or misdeployment could lead to unsafe behavior if models are applied outside their intended operating conditions or without adequate testing. We mitigate this by explicitly evaluating under heavy-tailed penalties and reporting risk-sensitive metrics, and by encouraging practitioners to validate performance and safety in their target environments before deployment.

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

## A. Limitations and Future Directions

Practical remedies include adding a distilled *one-step* RL head to avoid recursive backprop, and using score- or energy-weighted objectives that reduce full-path backprop for diffusion/flow policies (Koirala & Fleming, 2026; Park et al., 2025); both are natural directions for future improvement. On the critic side, modeling the *return distribution* itself with diffusion/flow value networks is a natural extension that may improve tail calibration without sacrificing actor expressiveness (Agrawalla et al., 2026; Zhang et al., 2025a).

Our theory is deliberately scoped to proposition-level guidance (e.g., KL–OOD relations) and does not provide finite-sample, high-probability guarantees under function approximation or partial coverage. Moreover, our primary evaluation uses CVaR as a direct objective for offline lower-tail risk control. While RAMAC can incorporate alternative distortion-based objectives, as illustrated by the Wang and CPW results in App. E.3.2, we do not study broader risk formulations such as mean–variance or entropic risk, nor explicit cost-constraint or barrier-based safety objectives; these constitute complementary directions to the setting considered here.

Promising future directions include strengthening the analysis to obtain non-asymptotic guarantees, exploring alternatives to BC (e.g., $f$-divergences or Wasserstein objectives) to trade off mode-covering versus mode-seeking behavior, extending the framework to dynamic or spectral risk measures and partial observability, connecting lower-tail risk optimization with explicit constrained-safety formulations, and developing *risk-aware offline-to-online fine-tuning* that preserves on-manifold exploration.

## B. Proofs

### B.1. Proof of Lemma 4.1

*Proof.* By assumption, $\pi_{\text{anch}}(\cdot \mid s)$ has a density $p(a \mid s)$ satisfying $p(a \mid s) \geq c > 0$ on the measurable leakage region $A_s \subseteq O_s$. Thus

$$\delta_s(\pi_{\text{anch}}) = \int_{O_s} p(a \mid s)\, da \ \geq \ \int_{A_s} p(a \mid s)\, da \ \geq \ c \cdot \lambda(A_s) > 0.$$

$\square$

#### B.1.1. Geometric Intuition Behind Lemma 4.1

Lemma 4.1 formalizes the geometric intuition that when the perturbation ball centered at $b$ overlaps the complement of the support, a residual policy can assign off-support probability mass; such leakage persists whenever the induced policy maintains nontrivial density on the overlapping region. The following points provide an intuitive view:

- **Thin support near $b$:** When the local margin $m(b)$ is small, any ball $B_\Phi(b)$ with $\Phi \gtrsim m(b)$ necessarily overlaps $O_s$, so some residual updates produce OOD actions even if $\|\zeta_\psi\| \leq \Phi$.

- **Nonconvex support:** If $\mathcal{S}_G(s)$ is nonconvex (e.g., a ring/annulus), anchors can sit near holes or concavities, again yielding small $m(b)$ and forcing overlap between $B_\Phi(b)$ and $O_s$.

- **Gradients pushing off the data surface:** The residual $\zeta_\psi(s, b)$ is trained to increase $Q$ or CVaR and is not constrained to remain tangent to the manifold, so gradients can push along the normal direction toward the boundary of $B_\Phi(b)$, further amplifying leakage when $m(b) \leq \Phi$.

Even when a coherent risk objective such as CVaR is used to train $\zeta_\psi$, Lemma 4.1 shows that OOD mass persists whenever the induced policy maintains nontrivial density on an off-support portion of its feasible perturbation region. Thin or nonconvex supported regions make this condition more likely.

### B.2. Proof of Lemma 4.2

*Proof.* Fix a state $s$ and define the $\beta$–support $I_s := \{a : \beta(a \mid s) > 0\}$ and its complement $O_s := I_s^c$. Assume $\beta \ll \pi_\theta$ on $I_s$ (so $\pi_\theta(I_s \mid s) > 0$ and the forward KL is finite).

Since $\beta(\cdot \mid s)$ has all its mass on $I_s$,

$$\mathrm{KL}(\beta\|\pi_\theta) = \int_{\mathcal{A}} \beta(a \mid s) \log \frac{\beta(a \mid s)}{\pi_\theta(a \mid s)} \, da = \int_{I_s} \beta(a \mid s) \log \frac{\beta(a \mid s)}{\pi_\theta(a \mid s)} \, da. \tag{1}$$

Define the normalization of $\pi_\theta$ to $I_s$:

$$\pi_I(a \mid s) := \pi_\theta(a \mid s, \, a \in I_s) = \frac{\pi_\theta(a \mid s)}{\pi_\theta(I_s \mid s)}, \qquad a \in I_s,$$

so that on $I_s$ we have the identity $\pi_\theta(a \mid s) = \pi_\theta(I_s \mid s) \, \pi_I(a \mid s)$.

Substitute the above factorization into (1) and use $\log(xy) = \log x + \log y$:

$$\log \frac{\beta(a \mid s)}{\pi_\theta(a \mid s)} = \log \frac{\beta(a \mid s)}{\pi_\theta(I_s \mid s) \, \pi_I(a \mid s)} = \log \frac{\beta(a \mid s)}{\pi_I(a \mid s)} - \log \pi_\theta(I_s \mid s). \tag{2}$$

Plug (2) into (1) and split the integral:

$$\mathrm{KL}(\beta\|\pi_\theta) = \int_{I_s} \beta(a \mid s) \log \frac{\beta(a \mid s)}{\pi_I(a \mid s)} \, da - \int_{I_s} \beta(a \mid s) \log \pi_\theta(I_s \mid s) \, da$$
$$= \underbrace{\mathrm{KL}\big(\beta(\cdot \mid s) \, \| \, \pi_I(\cdot \mid s)\big)}_{\geq 0} - \log \pi_\theta(I_s \mid s) \underbrace{\int_{I_s} \beta(a \mid s) \, da}_{=1}. \tag{3}$$

Here the last equality uses that $\log \pi_\theta(I_s \mid s)$ is constant in $a$, and that $\beta$ places total mass $1$ on $I_s$.

From (3) and nonnegativity of KL,

$$\mathrm{KL}(\beta\|\pi_\theta) \geq -\log \pi_\theta(I_s \mid s).$$

Exponentiating both sides gives

$$e^{-\mathrm{KL}(\beta\|\pi_\theta)} \leq \pi_\theta(I_s \mid s).$$

Since $\pi_\theta(I_s \mid s) = 1 - \delta_s(\pi_\theta)$ with $\delta_s(\pi_\theta) := \pi_\theta(O_s \mid s)$, we obtain the per–state OOD bound

$$\delta_s(\pi_\theta) \leq 1 - \exp\big\{-\mathrm{KL}(\beta(\cdot \mid s)\|\pi_\theta(\cdot \mid s))\big\}. \qquad \square$$

### B.3. Proof of Proposition 4.3 and Corollary 4.4

We first recall two standard facts. (i) *Dual representation of lower-tail CVaR.* For $\alpha \in (0, 1]$ and any integrable random variable $X$,

$$\mathrm{CVaR}_\alpha(X) = \inf_{w \in \mathcal{W}_\alpha} \mathbb{E}[wX], \qquad \mathcal{W}_\alpha := \Big\{w : 0 \leq w \leq \tfrac{1}{\alpha}, \, \mathbb{E}[w] = 1\Big\}. \tag{19}$$

(See, e.g., standard references on coherent risk measures / expected shortfall.)

(ii) *Coupling characterization of total variation.* For distributions $p, q$ on $\mathcal{A}$,

$$\mathrm{TV}(p, q) = \inf_{\gamma \in \Gamma(p,q)} \mathbb{P}_\gamma(A \neq A'), \tag{20}$$

where $\Gamma(p, q)$ denotes the set of all couplings $\gamma$ whose marginals are $A \sim p$ and $A' \sim q$.

*Proof of Proposition 4.3.* Fix $s$ and let $A \sim p(\cdot \mid s)$ and $A' \sim q(\cdot \mid s)$ be coupled by an arbitrary coupling $\gamma \in \Gamma(p, q)$. Define

$$X := X_s(A), \qquad Y := X_s(A').$$

Since $X_s : \mathcal{A} \to [m, M]$, we have $X, Y \in [m, M]$ almost surely.

**Step 1: CVaR is $(1/\alpha)$-Lipschitz in $L_1$.** Let $w_Y \in \arg\min_{w \in \mathcal{W}_\alpha} \mathbb{E}[wY]$ be a minimizer in (19) for $Y$ (if not attained, take an $\varepsilon$-minimizer and let $\varepsilon \to 0$). Then,

$$
\begin{aligned}
\mathrm{CVaR}_\alpha(X) - \mathrm{CVaR}_\alpha(Y) &= \inf_{w \in \mathcal{W}_\alpha} \mathbb{E}[wX] - \inf_{w \in \mathcal{W}_\alpha} \mathbb{E}[wY] \\
&\leq \mathbb{E}[w_Y X] - \mathbb{E}[w_Y Y] = \mathbb{E}[w_Y(X - Y)] \\
&\leq \mathbb{E}[w_Y|X - Y|] \leq \tfrac{1}{\alpha}\,\mathbb{E}[|X - Y|].
\end{aligned}
\tag{21}
$$

Swapping the roles of $X$ and $Y$ yields the reverse inequality, hence

$$
\big|\mathrm{CVaR}_\alpha(X) - \mathrm{CVaR}_\alpha(Y)\big| \leq \tfrac{1}{\alpha}\,\mathbb{E}[|X - Y|].
\tag{22}
$$

**Step 2: Bound $\mathbb{E}|X - Y|$ by TV via coupling.** Since $X_s$ is deterministic given the action input, $A = A'$ implies $X_s(A) = X_s(A')$. Thus,

$$
|X - Y| = |X_s(A) - X_s(A')| = |X_s(A) - X_s(A')|\mathbf{1}_{\{A \neq A'\}} \leq (M - m)\mathbf{1}_{\{A \neq A'\}},
$$

and taking expectation under the coupling $\gamma$ gives

$$
\mathbb{E}_\gamma[|X - Y|] \leq (M - m)\,\mathbb{P}_\gamma(A \neq A').
\tag{23}
$$

Combining (22) and (23),

$$
\big|\mathrm{CVaR}_\alpha(X_s(A)) - \mathrm{CVaR}_\alpha(X_s(A'))\big| \leq \tfrac{M-m}{\alpha}\,\mathbb{P}_\gamma(A \neq A').
$$

Since $\gamma$ was arbitrary, we can minimize over all couplings and apply (20):

$$
\big|\mathrm{CVaR}_\alpha(X_s(A)) - \mathrm{CVaR}_\alpha(X_s(A'))\big| \leq \tfrac{M-m}{\alpha}\,\mathrm{TV}\big(p(\cdot \mid s), q(\cdot \mid s)\big).
\tag{24}
$$

This proves the first inequality.

**Step 3: Pinsker.** Using Pinsker's inequality and symmetry of TV,

$$
\mathrm{TV}(p, q) = \mathrm{TV}(q, p) \leq \sqrt{\tfrac{1}{2}D_{\mathrm{KL}}\big(q(\cdot \mid s) \,\|\, p(\cdot \mid s)\big)},
$$

and substituting into (24) yields the second inequality. □

*Proof of Corollary 4.4.* Apply Proposition 4.3 with $p(\cdot \mid s) = \pi_\theta(\cdot \mid s)$ and $q(\cdot \mid s) = \beta(\cdot \mid s)$. Then (24) plus Pinsker gives

$$
\big|\mathrm{CVaR}_\alpha^\beta - \mathrm{CVaR}_\alpha^{\pi_\theta}\big| \leq \tfrac{M-m}{\alpha}\sqrt{\tfrac{1}{2}D_{\mathrm{KL}}\big(\beta(\cdot \mid s) \,\|\, \pi_\theta(\cdot \mid s)\big)}.
$$

□

# C. Related Work

We review works most relevant to our *risk-aware generative trajectory* view—policies that map noise to actions through a differentiable path and how safety is enforced therein—while avoiding repetition of the core background already covered in the main text. For a broad taxonomy of offline RL, see (Prudencio et al., 2023).

**Expressive Generative Policies** The main paper reviews diffusion and flow-matching policies (e.g., DiffusionQL, IDQL, FQL). Here we note complementary developments not detailed there: (i) *Diffusion-policy imitation learning with DDIM-style fast sampling* that accelerates inference while retaining diffusion expressiveness in visuomotor or robot control (Ankile et al., 2024; Chi et al., 2025); (ii) on the flow-matching side, recent generative policies for robot manipulation replace diffusion with flow matching to achieve similar expressiveness with faster, more stable inference (Gao et al.; Yan et al., 2025; Zhang et al., 2025b); (iii) *transformer-based trajectory and policy models*, including trajectory- and sequence-level generative decision-makers such as Decision Diffuser and Decision Transformer, and hierarchical visuomotor transformers for bimanual or gaze-conditioned manipulation (Ajay et al., 2023; Chen et al., 2021; Chuang et al., 2026; Janner et al., 2022; Lee et al., 2024). These works bolster the case for expressive generative decision-making models but remain *risk-neutral* in objective design.

**Risk-Sensitive RL** Beyond expectation-oriented objectives, risk-sensitive control formalizes tail-aware criteria via coherent/dynamic risk measures for MDPs (Ruszczyński, 2010). Among coherent measures, CVaR admits sampling- and policy-gradient formulations suitable for RL (Tamar et al., 2015a;b), and has been linked to robustness via CVaR–robust trade-offs (Chow et al., 2015). In the *offline* regime, safety is often operationalized as high-confidence off-policy evaluation/improvement from fixed logs (Laroche et al., 2019; Thomas et al., 2015), which bound deployment risk yet do not address how *expressive generators* should receive lower-tail gradients.

**Mixture-Policy CVaR Optimization.** Luo et al. (Luo et al., 2024) propose a mixture policy parameterization for CVaR optimization in online RL, where a risk-neutral policy and an adjustable component are combined to form a risk-averse policy and improve the sample efficiency of CVaR policy gradients. While both their work and ours aim to shape the lower tail of returns, their method uses standard parametric actors and does not address offline data, generative policies, or explicit constraints on out-of-distribution actions. By contrast, RAMAC targets offline risk-sensitive control with a single expressive generative actor (diffusion/flow), a distributional critic, a BC+CVaR objective, an analysis of per-state OOD-action suppression, and hazard relabeling.

**Closest Lines and Delineation** Concurrent actor–critic lines that couple diffusion with value learning remain expectation-oriented:(Zhang et al., 2025a) stabilizes *online* diffusion actors with distributional critics and double-$Q$ but does not backpropagate CVaR along the denoising path;(Fang et al., 2025) formulates *offline* constrained policy iteration as diffusion noise regression under KL/BC regularization; a distributional SAC variant (Ma et al., 2025) improves risk sensitivity via value-law estimation and distorted expectations (e.g., CVaR), but operates with standard unimodal Gaussian policies in an online setting, rather than shaping a multimodal diffusion/flow actor under behavior-regularized offline constraints; the diffusion-policy instantiation (Liu et al., 2025) targets multi-modality but likewise reports no CVaR along the multi-step generation. Risk-averse offline methods relying on behavior priors (e.g., (Urpí et al., 2021)), and diffusion-prior (e.g., (Chen et al., 2025)) use anchor–perturb/mixing mechanisms, while (Ma et al., 2021) imposes conservative distributional critics (value pessimism). These approaches either (i) optimize expectation-oriented objectives with expressive generators or (ii) control risk via mixing or pessimism, in contrast to our distributional risk shaping without anchor mixing.

# D. Implementation Details

**Actor Architecture** RAMAC employs a reparameterized generative actor $a = \psi_\theta(s, z)$ so that gradients from the risk term flow through the entire generative trajectory. RADAC instantiates $\psi_\theta$ as a denoising diffusion policy with VP schedule and $T=5$ denoising steps; the score network is an MLP (hidden 256–256, SiLU) following (Wang et al., 2023). RAFMAC instantiates $\psi_\theta$ as a deterministic flow–matching ODE solved by Euler with *flow_steps* $K=10$; the velocity field is an MLP (hidden 512–512, SiLU) following (Park et al., 2025). For both, the actor objective is $\mathcal{L}_\pi = \mathcal{L}_{\text{BC}} + \eta \, \mathcal{L}_{\text{Risk}}$ (as shown in Eq. 11), where $\mathcal{L}_{\text{BC}}$ is the model's native BC loss (denoising loss for diffusion and flow-matching loss for flow), serving as the surrogate counterpart of Eq. 10, and $\mathcal{L}_{\text{Risk}} = -\mathbb{E}_{s,a\sim\pi_\theta}[\text{CVaR}_\alpha(Z_\phi(s,a))]$ with $\alpha=0.1$ as shown in Eq. 9.

**Distributional Critic Architecture** As described in Sec. 3.1, both RAMAC (diffusion actor) and RAFMAC (flow actor) use a Double IQN critic $Z_\phi(s, a; \tau)$ that returns the $\tau$-quantile of the return distribution. We instantiate two critics $Z_{\phi_1}, Z_{\phi_2}$ and train them with the quantile Huber loss ($\kappa=1$), using a standard double-critic scheme where targets are formed with a minimum over a slowly updated target copy $Z_{\bar\phi}$ to curb overestimation.

For a batch $(s, a, r, s') \sim \mathcal{D}$, the bootstrap action is always produced by the same generative actor used in the policy loss:

$$a' = \psi_\theta(s', z'), \qquad z' \sim \mathcal{N}(0, I),$$

so that the critic learns return quantiles under the current actor and, in turn, supplies tail-sensitive gradients (via CVaR) to the risk-aware actor update (Step 2, Sec. 3.2). Concretely, one can view the temporal-difference (TD) objective as approximating expectations over $\tau, \tau' \sim \mathcal{U}(0, 1)$, while the actor-side CVaR objective uses $\tau \sim \mathcal{U}(0, \alpha)$.

In implementation, we replace stochastic quantile sampling with midpoint quadrature on fixed uniform grids. This provides a deterministic approximation to the relevant quantile integrals and removes Monte Carlo sampling variance for a fixed critic. Instead of drawing $\tau$ at random, we use

$$\mathcal{T}_N = \left\{ \tau_i = \frac{i-\frac{1}{2}}{N} \right\}_{i=1}^{N}, \tag{25}$$

and approximate $\mathrm{CVaR}_\alpha$ by averaging the lowest-$\alpha$ fraction of quantiles. For CVaR at level $\alpha$, let $m = \lfloor \alpha N \rfloor$; then

$$\mathrm{CVaR}_\alpha\big(Z_\phi(s,a)\big) \approx \frac{1}{m} \sum_{i=1}^m Z_\phi\big(s,a;\tau_i\big), \qquad \tau_i \in \mathcal{T}_N. \tag{26}$$

This is equivalent in expectation to drawing $\tau \sim \mathcal{U}(0,\alpha)$ (Eq. 8), but the deterministic grid substantially reduces variance in both the critic loss and the actor's CVaR objective.

For the TD targets we use another grid $\mathcal{T}_{N'}$ and define the residual for each pair of source and target quantiles as

$$\delta_{\tau_i,\tau_j'} = r + \gamma\, Z_{\bar{\phi}}(s',a';\tau_j') - Z_\phi(s,a;\tau_i), \qquad \tau_i \in \mathcal{T}_N,\; \tau_j' \in \mathcal{T}_{N'}.$$

The critic minimizes the standard quantile Huber loss (Dabney et al., 2018; Rowland et al., 2019)

$$\mathcal{L}_\kappa(\delta;\tau) = \big| \tau - \mathbf{1}_{\{\delta<0\}} \big| \times \begin{cases} \frac{\delta^2}{2\kappa}, & |\delta| \le \kappa, \\ |\delta| - \frac{\kappa}{2}, & \text{otherwise}, \end{cases} \tag{27}$$

with $\kappa = 1$. Averaging over all $N \times N'$ quantile pairs yields the final critic objective

$$\mathcal{L}_{\mathrm{critic}}(\phi) = \mathbb{E}_{(s,a,r,s'),\,a'} \left[ \frac{1}{NN'} \sum_{i=1}^N \sum_{j=1}^{N'} \mathcal{L}_\kappa\big(\delta_{\tau_i,\tau_j'};\tau_i\big) \right]. \tag{28}$$

Thus Eq. 28 is exactly the compact critic objective in Sec. 3.1, with deterministic grids $(\tau_i,\tau_j')$ used in place of stochastic $(\tau,\tau')$. Optimising this loss yields an estimate of the return distribution; its lower tail is then aggregated into $\mathrm{CVaR}_\alpha$ to provide the tail-aware gradients used in the RAMAC actor loss (Step 2).

**Hyperparameters**  Unless noted, we use Adam for all networks (default $3 \times 10^{-4}$), batch size 256, discount $\gamma = 0.99$, soft target update $\tau_{\mathrm{target}} = 0.005$, and no LR decay. RAMAC's specific hyperparameters (critic LR, IQN size, $\eta$, gradient–norm clipping, optional $Q$–target clipping, etc.) are listed in Table 7.

**RAFMAC Risk-Weight Tuning**  For RAFMAC, we swept $\eta \in \{1, 10, 50, 100, 300, 1000\}$ and *unified to $\eta=1000$* for all datasets; critic settings are fixed ($lr_{\mathrm{critic}} = 3 \times 10^{-4}$, `emb_dim`= 128, `n_quantiles`= 32) (Table 7).

**Critic–Target Clipping**  Where specified, target returns are clipped ($[-300, 300]$ or $[-150, 150]$) to dampen rare outliers without affecting on–manifold learning.

# E. Additional Experimental Results

### E.1. More 2D Synthetic Task Results

**Behavior cloning task Fig. 5**  On the 2D bandit dataset, three BC models show generator-specific patterns. CVAE-BC collapses topology and places probability in the low-density gap. Diffusion-BC most faithfully reproduces both the ring and the inner cluster with appropriate thickness. Flow-Matching BC draws a sharp ring but allocates less mass to the center and shows edges spread slightly outward. These baselines confirm that a suitably trained generative model can represent the full multimodal support of the dataset.

**RADAC dynamics over training Fig. 6**  RADAC starts from the same diffusion generator that is first fit to the behavior data, and then applies CVaR-based policy updates on top of the BC objective. In Fig. 6 we fix the RL weight $\eta$ to the value used in Fig. 3 and visualize how the induced policy evolves over training.

Early in training $\sim$200 epochs, the critic has not yet identified the outer ring as hazardous: high-mean returns on the ring dominate the low-quantile signal, so the CVaR-guided updates still allocate substantial mass to the high-return ring while also covering the safe central cluster. As learning proceeds and low-quantile returns on the ring are observed, CVaR increasingly penalizes ring actions; between about 400 and 800 epochs the ring thins and probability mass shifts inward as the central cluster grows. By roughly 950 epochs most mass concentrates near the safe center, matching the final snapshot in Fig. 3.

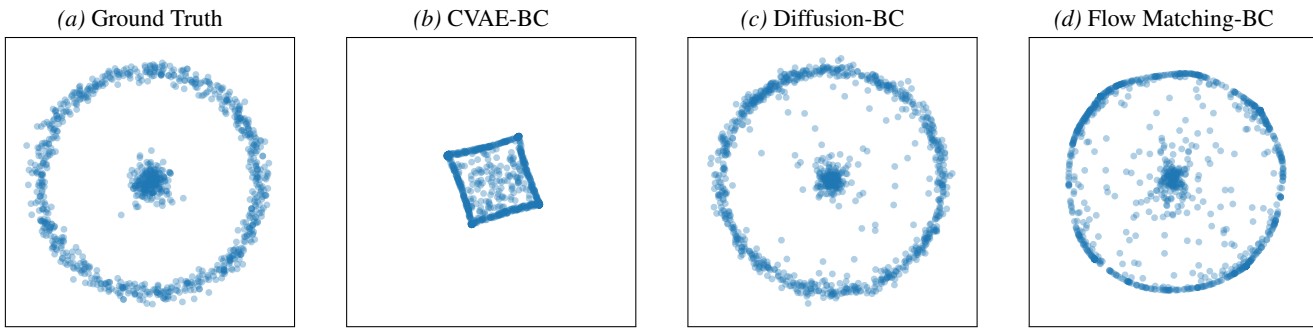

*(a)* Ground Truth      *(b)* CVAE-BC      *(c)* Diffusion-BC      *(d)* Flow Matching-BC

*Figure 5.* **Behavior cloning on the Toy Risky Bandit dataset.** Each panel shows i.i.d. samples from the BC Policy. CVAE-BC mixes modes and places points in the low-density gap; Diffusion-BC reproduces both the outer ring and the central cluster; Flow-Matching BC yields a crisp ring but assigns less mass to the center.

This trajectory highlights that RADAC uses the expressive diffusion policy to first capture the multimodal behavior distribution and then *reallocate* mass toward the high-CVaR central mode once its tail risk is correctly estimated, rather than collapsing due to limited capacity. The explicit BC-risk trade-off as the RL weight $\eta$ varies is further analyzed in Fig. 9.

**On conservatism and hyperparameters in prior baselines.** The same Risky–Bandit geometry also helps explain why conservative tuning of prior methods (DiffusionQL, FQL, ORAAC-style anchor-perturbation) does not eliminate their structural failure modes. As defined in App. F.1, behavior actions lie on a thin, high-mean but heavy-tailed outer ring and a lower-mean but light-tailed central Gaussian cluster. Geometrically, each behavior anchor $b$ lies on a narrow manifold with margin $m(b)$ to the low-density gap and to regions that are unobserved in the dataset. Once a perturbation ball $\mathbb{B}_\varepsilon(b)$ has $\varepsilon \gtrsim m(b)$, it necessarily crosses into these off-support regions. In this regime, residual or anchor-perturbation policies are not constrained to stay tangent to the data manifold, so even small residuals can systematically produce OOD actions, which in turn leads to unstable Q-estimates and higher exposure to the trap-heavy ring. This is precisely the behavior illustrated in Fig. 3(f-g).

For DiffusionQL and FQL, the RL term is always driven by a risk-neutral Q-function. Sweeping the RL scaling coefficient $\eta$ (as shown in Fig. 7) shows the expected dichotomy: as $\eta \to 0$ the policies revert to BC-like behavior, recovering the multimodal density of the behavior data; as $\eta$ increases, mass increasingly concentrates on the high-mean ring, despite its heavy lower tail. FQL's flow-matching prior retains a bit more mass near the center, but the dominant mode still tracks the risky ring. No choice of $\eta$ can simultaneously prevent this tendency and retain a nontrivial RL component, because the objective is fundamentally risk-neutral.

Fig. 8 shows ORAAC-style anchor-perturbation exhibits an analogous tradeoff. When the risk-distortion weight $\lambda$ is set to zero, the method again collapses to BC on the behavior manifold. As $\lambda$ increases, the residual pushes probability mass away from the BC solution: in the Risky–Bandit geometry this either splits modes or routes probability through low-density regions between them, and for larger $\lambda$ the policy can "jump" toward arbitrary corners of the action space, far from any behavior support (as predicted by the analysis in Sec. 4).

In other words, the observed failures of Fig. 3(c-g) are not artifacts of insufficient hyperparameter tuning but a structural consequence of combining risk-neutral or anchor-perturbation objectives with thin, non-convex data supports. RADAC replaces this with an on-manifold, risk-aware update that reduces tail risk without relying on off-support residuals.

For completeness, we also sweep the RL weight $\eta$ in RADAC on the same Risky–Bandit (Fig. 9). When $\eta \to 0$, the objective reduces to pure BC and the diffusion policy faithfully matches the multimodal behavior distribution, reproducing both the outer ring and the central cluster. As $\eta$ increases, the CVaR-guided term gradually reallocates probability mass away from the high-mean but heavy-tailed ring toward the light-tailed central mode while staying on-support: the ring thins and eventually disappears, leaving a compact cluster around the safe center. Unlike DiffusionQL/FQL and ORAAC-style anchor–perturbation, this sweep does not create spurious off-support modes or low-density bridges; instead, it yields a smooth trade-off between ring coverage and tail-risk reduction, consistent with the on-manifold, risk-aware behavior-regularized objective in Eq. 11.

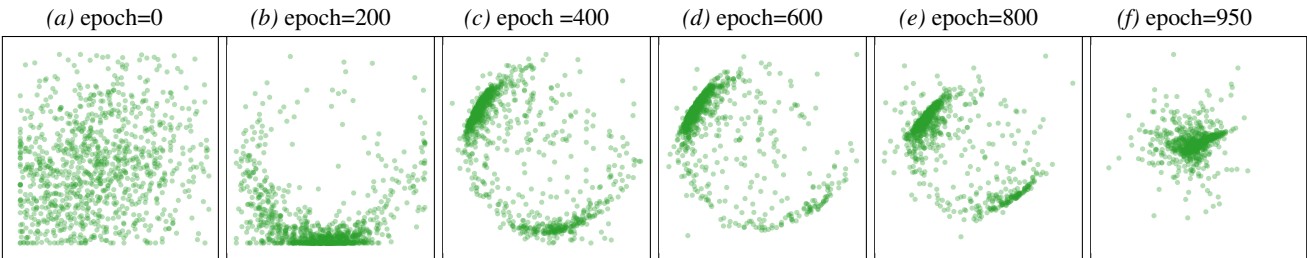

*Figure 6.* **RADAC dynamics on the toy task.** Mass gradually moves from the risky ring to the safe center: the ring thins (400–800 epochs) and the central cluster grows, ending with most mass at the center ( 950 epochs). BC keeps the policy on-manifold while CVaR reduces lower-tail risk.

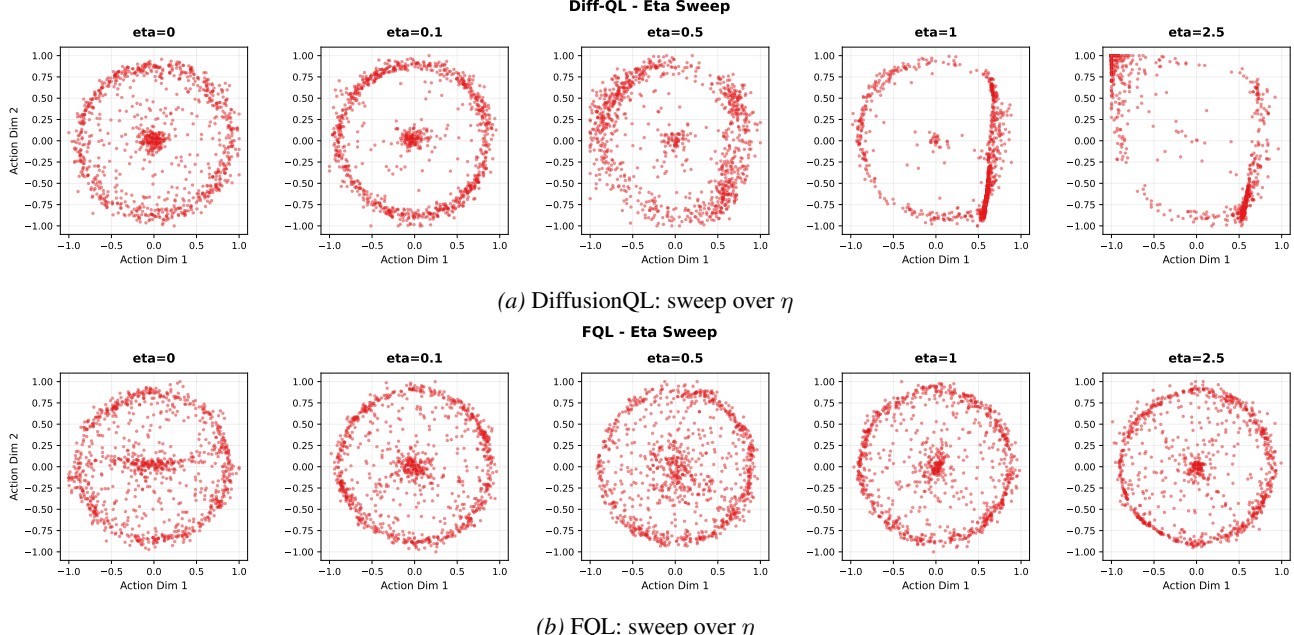

*Figure 7.* **Effect of RL weight $\eta$ in Toy Risky Bandit.** Each panel shows a policy trained on the 2D Toy Risky Bandit with a different RL scaling coefficient $\eta$. As $\eta \to 0$, especially DiffusionQL reverts to BC-like behavior and recovers the multimodal behavior density. As $\eta$ increases, probability mass concentrates on the high-mean outer ring despite its heavy lower tail, illustrating the risk-neutral tendency discussed in Sec. 3 and App. E.

### E.2. Extended Stochastic-D4RL Results

**Protocol**    Table 3 reports an additional fixed-checkpoint evaluation under a uniform 1000-step rollout horizon for all included methods and tasks. This evaluation is complementary to the main-text comparison: it removes checkpoint-selection effects and therefore should not be interpreted as reproducing the exact ranking in Table 1.

**Consistency with the Main-Text Trends**    The fixed-checkpoint evaluation preserves several strengths of RADAC, particularly on HALFCHEETAH-m-e and HOPPER-m-e, while also revealing settings in which FQL or RAFMAC achieves stronger mean or lower-tail performance.

**Pareto Frontier Analysis: Return vs. Safety Violations**    Figure 10 plots mean return against the number of safety violations per episode, with each point corresponding to an evaluation snapshot collected during training and color indicating the training epoch. This trajectory-level view complements the fixed 1000-step comparison in Table 3. We organize the discussion by algorithm.

Across datasets, RADAC populates the upper-left region of the frontier: for comparable return, it tends to incur fewer violations. Only for HALFCHEETAH–medium–expert, RADAC sometimes drifts up-right (higher return with slightly more

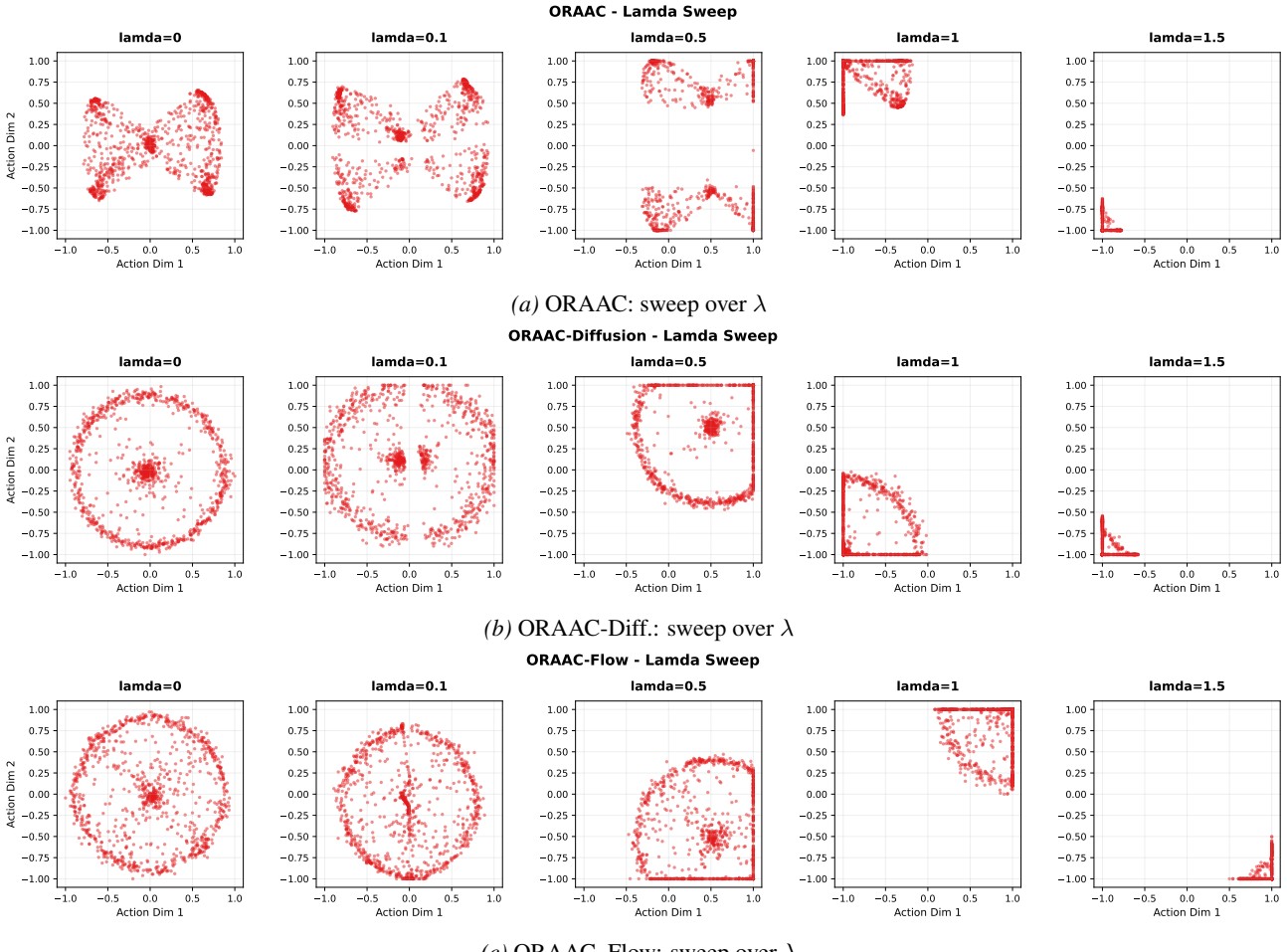

*Figure 8.* **Effect of risk weight $\lambda$ in anchor-perturbation baselines.** Each panel visualizes policies on the 2D Toy Risky Bandit as the risk-distortion weight $\lambda$ is swept. For $\lambda = 0$, all ORAAC variants reduce to BC on the behavior manifold. As $\lambda$ increases, residual updates push probability mass away from the BC solution: modes can split or leak through low-density gaps, and for larger $\lambda$ the policy may jump toward regions far from any behavior support, consistent with the analysis in Sec. 4.

violations) because the penalty is light and non-terminating, so near-threshold speed pays off, consistent with its best Mean/CVaR. Mechanistically, diffusion with CVaR guidance enables fine-grained reweighting away from safety thresholds while BC keeps samples on-manifold, so trajectories in the plot drift left (fewer violations) without sacrificing return. ORAAC forms the frontier on HOPPER-m-e with few violations and strong returns, matching its leading scores under terminating pose hazards. In other settings it remains reliably conservative (low violations) at the cost of mean on some tasks, consistent with anchor-based regularization. FQL often achieves high-mean points but with comparatively higher violation counts in the Pareto plot. Without tail-aware guidance, safety depends on the expected-value critic and task smoothness, explaining the variability across datasets. DiffusionQL exhibits wider scatter: runs either reach moderate returns with elevated violations or collapse to low-return, near-zero violation regions. This variability is consistent with value-only guidance under stochastic penalties and matches its weaker CVaR. CODAC clusters in the low-return/low-violation corner across tasks, as expected from conservative critics.

**Additional Qualitative Safety Analysis on MEDIUM–EXPERT** Fig. 4 in the main text reports qualitative safety heatmaps on the MEDIUM-REPLAY datasets. Here we provide the corresponding plots on MEDIUM-EXPERT to verify that the same qualitative trends hold under a different data regime. As in the main text, shaded bands indicate the risk-free operational ranges (HalfCheetah: $v \leq 10$; Walker2d: $|\theta| \leq 0.5$; Hopper: $|\theta| \leq 0.1$). Across tasks in Fig. 11, **RADAC** concentrates probability mass inside or near the risk-free boundary while preserving high-reward behavior supported by the dataset. In contrast, **DiffusionQL** tends to under-explore near the boundary due to risk-blind bootstrapping under rare hazards, often

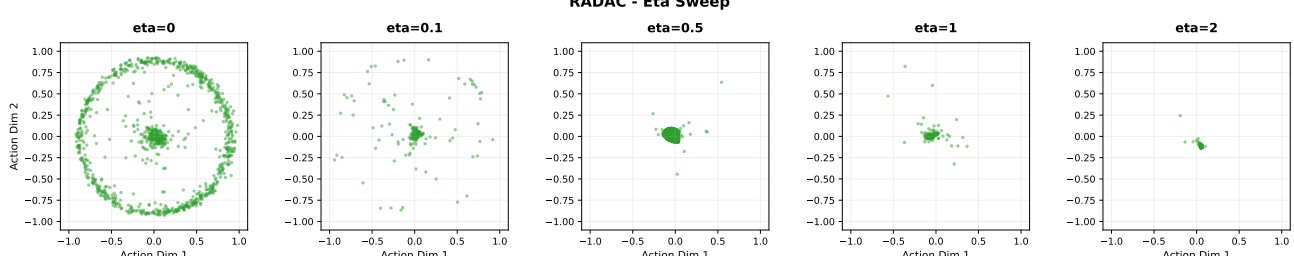

*Figure 9.* **Effect of RL weight $\eta$ in RADAC.** As $\eta$ increases from 0 (pure BC), probability mass smoothly moves from the risky outer ring to the safe central cluster, without creating off-support modes.

yielding overly conservative action distributions. Finally, **ORAAC** (prior-anchored perturbation) can allocate nontrivial mass near low-density regions induced by the anchored geometry, consistent with the leakage intuition in Sec. 4. Overall, the MEDIUM–EXPERT results are consistent with the main-text conclusion.

### E.3. Ablation Study

#### E.3.1. OBJECTIVE ABLATION: ROLES OF BC AND CVAR

To isolate the roles of behavior regularization and tail-risk optimization, we compare the full RADAC objective against BC-only and CVaR-only variants on two representative Stochastic-D4RL tasks. We additionally report the state-conditioned 1-NN OOD action rate used in Sec. 5.3. The results support a two-stage interpretation: BC substantially suppresses OOD leakage, while the CVaR term improves lower-tail performance within this low-OOD regime.

#### E.3.2. CHOICE OF RISK DISTORTION

We evaluate RADAC and RAFMAC with three risk distortions CVaR, Wang, and CPW under the same 1000-step evaluation protocol used above. Across seeds, Wang generally tilts updates toward higher means and weaker tails; CPW sits between CVaR and Wang but shows higher variance across seeds. Overall, CVaR is the most reliable choice for lower-tail control at comparable mean.

#### E.3.3. EFFECT OF THE TAIL SAMPLE SIZE $K$ AND $N$ ON THE CVAR ESTIMATOR

To clarify the role of the tail sample size $K$ in our IQN-based CVaR estimator, we run an experiment on a subset of D4RL, HALFCHEETAH-MEDIUM-REPLAY-V2 and WALKER2D-MEDIUM-REPLAY-V2. During training we randomly sample 2,000 states from the replay buffer and store them as a fixed evaluation set $\{s_j\}_{j=1}^{2000}$. For each state $s_j$, each $K \in \{2, 4, 8, 16\}$, and each risk level $\alpha \in \{0.05, 0.1, 0.2\}$ we estimate $\text{CVaR}_\alpha$ using the offline-selected RADAC policy (trained with $\alpha_{\text{train}} = 0.1$).

Given a state $s$, we draw $K$ i.i.d. pairs $(a_k, \tau_k)$ with $a_k \sim \pi_\theta(\cdot \mid s)$ and $\tau_k \sim \text{Unif}(0, \alpha)$, evaluate both distributional critics $Z_{\phi_1}, Z_{\phi_2}$, and form the tail-sampling CVaR estimator

$$\widehat{\text{CVaR}}_\alpha(s) = \frac{1}{K} \sum_{k=1}^{K} \min_{i \in \{1,2\}} Z_{\phi_i}(s, a_k; \tau_k). \tag{29}$$

For each $(K, \alpha)$ we repeat this procedure 100 times per state (using new $(a_k, \tau_k)$ draws each time), compute the variance of $\widehat{\text{CVaR}}_\alpha(s_j)$ across the 100 repetitions for each state $s_j$, and then aggregate these per-state variances by their mean and standard deviation over $j = 1, \ldots, 2000$.

Figure 12 reports the resulting *estimator variance* as a function of $K$ for $\alpha \in \{0.05, 0.1, 0.2\}$. Across both tasks and all three risk levels the variance decreases approximately at a $1/K$ rate: using only $K = 2$ tail samples yields noisy CVaR estimates, while increasing $K$ to 4 and 8 substantially reduces the variance. The marginal improvement from $K = 8$ to $K = 16$ is much smaller, despite doubling the number of tail samples. This supports our choice of a moderate tail size (with $K$ in the range 8–16 across tasks in our experiments), which provides a good trade-off between estimator noise and computational cost. Note that the absolute variance scale differs between environments.

*Table 3.* Stochastic-D4RL (1000-step evaluation): Mean and $\text{CVaR}_{0.1}\pm$ s.e. over 5 seeds.

| Environment, Dataset | Algorithm | Mean | CVaR |
|---|---|---|---|
| HalfCheetah-m-e | CQL | $-0.97\pm0.24$ | $-2.24\pm0.43$ |
| | CODAC | $-0.12\pm0.08$ | $-1.48\pm0.27$ |
| | ORAAC | $4106.25\pm177.48$ | $3692.79\pm466.31$ |
| | FQL | $4695.46\pm65.97$ | $4025.12\pm230.08$ |
| | DiffusionQL | $-118.72\pm64.53$ | $-198.01\pm76.76$ |
| | RAFMAC | $5084.12\pm230.43$ | $3735.37\pm827.60$ |
| | RADAC | $5659.40\pm131.94$ | $4667.96\pm42.59$ |
| Walker2d-m-e | CQL | $-10.32\pm6.27$ | $-73.38\pm9.02$ |
| | CODAC | $27.56\pm6.26$ | $-35.30\pm15.36$ |
| | ORAAC | $663.23\pm181.31$ | $205.21\pm65.45$ |
| | FQL | $2457.68\pm208.80$ | $448.48\pm208.81$ |
| | DiffusionQL | $-32.33\pm4.59$ | $-68.43\pm11.28$ |
| | RAFMAC | $3567.89\pm206.63$ | $356.20\pm987.34$ |
| | RADAC | $2760.21\pm689.32$ | $322.76\pm757.44$ |
| Hopper-m-e | CQL | $43.22\pm29.48$ | $-65.90\pm36.42$ |
| | CODAC | $31.59\pm28.74$ | $-77.88\pm34.33$ |
| | ORAAC | $660.07\pm157.55$ | $400.84\pm142.60$ |
| | FQL | $393.64\pm27.75$ | $77.93\pm60.53$ |
| | DiffusionQL | $-38.75\pm27.68$ | $-212.49\pm91.99$ |
| | RAFMAC | $370.11\pm39.95$ | $-120.09\pm56.34$ |
| | RADAC | $1513.27\pm101.71$ | $967.21\pm160.68$ |
| HalfCheetah-m-r | CQL | $-38.85\pm38.44$ | $-40.23\pm38.44$ |
| | CODAC | $-0.12\pm0.08$ | $-1.48\pm0.26$ |
| | ORAAC | $315.87\pm69.27$ | $161.54\pm68.76$ |
| | FQL | $1909.57\pm395.55$ | $568.43\pm256.85$ |
| | DiffusionQL | $2261.16\pm531.18$ | $1439.77\pm461.28$ |
| | RAFMAC | $2696.61\pm110.68$ | $1499.80\pm394.08$ |
| | RADAC | $2674.72\pm51.76$ | $1401.03\pm199.08$ |
| Walker2d-m-r | CQL | $-14.68\pm5.52$ | $-95.30\pm18.50$ |
| | CODAC | $26.39\pm7.97$ | $-36.56\pm12.92$ |
| | ORAAC | $160.23\pm147.55$ | $-359.49\pm302.72$ |
| | FQL | $647.33\pm166.12$ | $-29.64\pm110.73$ |
| | DiffusionQL | $-23.50\pm4.44$ | $-53.55\pm12.30$ |
| | RAFMAC | $778.00\pm130.03$ | $7.92\pm35.77$ |
| | RADAC | $383.87\pm288.95$ | $-309.70\pm246.62$ |
| Hopper-m-r | CQL | $2.28\pm42.17$ | $-130.48\pm53.25$ |
| | CODAC | $3.61\pm18.41$ | $-105.41\pm19.86$ |
| | ORAAC | $-30.00\pm32.77$ | $-179.92\pm61.46$ |
| | FQL | $448.26\pm70.39$ | $-33.21\pm43.38$ |
| | DiffusionQL | $-22.15\pm24.93$ | $-163.82\pm59.18$ |
| | RAFMAC | $350.36\pm33.05$ | $-36.69\pm28.35$ |
| | RADAC | $453.64\pm68.46$ | $-87.04\pm123.96$ |

**Role of the number of quantiles** $N$**.**  The tail estimator in Eq. 29 depends on the number of tail samples $K$, while the IQN critic itself is trained with $N$ quantile samples per state–action pair. In all experiments we use moderate values $N \in \{16, 32\}$ (see Table 7), which are standard in prior IQN-based work and make the critic updates stable. Increasing $N$ primarily reduces the variance of the critic update and smooths the learned value distribution, but once $N$ is in this moderate range we do not observe qualitative changes in the CVaR estimates or policy performance, while the computational cost grows roughly linearly in $N$. Hence we treat $N$ as a fixed architectural hyperparameter.

### E.3.4. OOD DETECTOR ROBUSTNESS

To check that the RADAC < ORAAC trend in Table 2 is not an artifact of the 1-NN score, we repeated the analysis with two additional detectors: a Local Outlier Factor (LOF) and a simple Mahalanobis detector based on a single Gaussian fit to the dataset joint $(s, a)$ points. Table 6 reports the three Stochastic-D4RL medium-expert tasks with these detectors. On all datasets, all three detectors agree with the main-text result and rank RADAC as having substantially lower OOD action rates than ORAAC variants, indicating that the RADAC < ORAAC ordering is robust to the detector choice. Implementation details of these detectors are provided in App. F.4

### E.4. Runtime and Inference Latency

We compare wall-clock training time and per-action inference latency for DiffusionQL, FQL, and RADAC on HOPPER-MEDIUM-EXPERT-V2. All methods use the same A6000 GPU, PyTorch/CUDA stack, and training budget. Training time is

*Table 4.* Objective ablation on representative Stochastic-D4RL tasks over 3 seeds. OOD denotes the state-conditioned 1-NN action rate (%); smaller is better.

| Dataset | Variant | Mean | CVaR$_{0.1}$ | OOD |
|---|---|---|---|---|
| HalfCheetah-m-r | RADAC (full) | 518.7 | 341.0 | 2.08 |
| | BC-only | 485.9 | 269.5 | 2.42 |
| | CVaR-only | 2.9 | $-28.3$ | 12.97 |
| Walker2d-m-r | RADAC (full) | 585.9 | 128.3 | 0.49 |
| | BC-only | 617.3 | $-95.6$ | 0.46 |
| | CVaR-only | $-14.4$ | $-53.0$ | 62.87 |

*Table 5.* Ablation (1000-step evaluation). RADAC/RAFMAC with CVaR, Wang, and CPW on HALFCHEETAH-Medium-Replay and WALKER2D-Medium-Replay. Scores are mean $\pm$ s.e. over 3 seeds.

| Method | Distortion | HalfCheetah–m–r | | Walker2d–m–r | |
|---|---|---|---|---|---|
| | | Mean | CVaR$_{0.1}$ | Mean | CVaR$_{0.1}$ |
| RADAC | CVaR | $2758.5 \pm 84.1$ | $1759.5 \pm 71.5$ | $681.3 \pm 409.3$ | $-395.1 \pm 438.3$ |
| RADAC | Wang | $2653.5 \pm 86.5$ | $310.8 \pm 92.6$ | $417.3 \pm 397.0$ | $-52.1 \pm 11.4$ |
| RADAC | CPW | $2777.9 \pm 93.7$ | $1061.6 \pm 731.7$ | $64.3 \pm 149.3$ | $-203.6 \pm 69.8$ |
| RAFMAC | CVaR | $2835.8 \pm 116.3$ | $1981.2 \pm 405.3$ | $698.8 \pm 215.5$ | $5.6 \pm 60.8$ |
| RAFMAC | Wang | $2625.6 \pm 113.8$ | $462.5 \pm 427.6$ | $552.2 \pm 134.8$ | $-706.4 \pm 687.5$ |
| RAFMAC | CPW | $2539.2 \pm 31.1$ | $95.9 \pm 92.3$ | $360.7 \pm 49.6$ | $-71.6 \pm 22.1$ |

measured between the first and last gradient update; latency is the average GPU time to produce a single action for $10^4$ replay states with `torch.no_grad`.

RADAC replaces the scalar critic with a distributional IQN and optimizes a CVaR objective, but this does not dominate runtime. In these methods, most of the computational cost comes from the expressive generative actor (diffusion or flow), not from the critic. The extra work required by IQN, evaluating a small number of quantiles and averaging the bottom $\alpha$-fraction for CVaR, adds only minor overhead relative to a full diffusion / flow pass.

Conversely, FQL carries its own overhead by training both a flow-matching prior and a distilled one-step policy head, and the DiffusionQL implementation uses slightly heavier hyperparameters (e.g., a larger actor) than RADAC. As a result, under our implementation and hyperparameter choices, RADAC ends up slightly faster in wall-clock time on HOPPER-MEDIUM-EXPERT-V2. We do not claim that RADAC is intrinsically faster than DiffusionQL or FQL; these measurements simply show that the CVaR + distributional critic extension does *not* introduce an order-of-magnitude runtime penalty. Inference latency is likewise dominated by the shared diffusion/flow backbone, and the RADAC actor achieves comparable or lower per-action latency than the risk-neutral expressive baselines when using similar numbers of denoising / flow steps.

### E.5. Return Distributions of Rollout Trajectories

We visualize the empirical return distributions under the stochastic hazard wrapper for HALFCHEETAH-MEDIUM-EXPERT-V2. For each method we aggregate 30–60 evaluation episodes across three seeds and plot histograms and kernel-density estimates of the rollout returns, together with vertical markers for the mean, median, and CVaR$_{0.1}$ (Fig. 14). On HALFCHEETAH-MEDIUM-EXPERT-V2, for example, RADAC concentrates mass in a high-return band (CVaR$_{0.1} \approx 4.7 \times 10^3$) while ORAAC exhibits a multimodal distribution with occasional near-zero or negative episodes, and DiffusionQL/CQL collapse near zero. Since hazard events in our wrapper correspond to large negative penalties without early termination, catastrophic hazard-inducing episodes populate the extreme left tail; the reduced tail mass for RADAC thus reflects a lower frequency of hazard-heavy trajectories rather than mere over-conservatism.

### E.6. Risk-Return Frontier under CVaR Level

To provide a clearer view of the safety-return trade-off, we visualize how the CVaR level $\alpha$ affects RADAC's behavior on HALFCHEETAH-MEDIUM-REPLAY-V2 and WALKER2D-MEDIUM-REPLAY-V2. For each environment, we train RADAC with $\alpha \in \{0.05, 0.10, 0.20\}$ using the same hyperparameters as in the main Stochastic-D4RL experiments, and aggregate results across multiple seeds. For each run, we select the checkpoint with the highest normalized score and compute the mean normalized return and empirical CVaR$_{\alpha}$ from eval rollouts under the risky wrapper. We then report the seed-averaged

*Table 6.* Detector–robust OOD action rates $\varepsilon_{\text{act}}$ (%, mean $\pm$ std over 3 seeds) on Stochastic–D4RL medium–expert tasks. Smaller is better.

| Env | Algo | 1-NN (state-cond.) | LOF $(s, a)$ | Mahalanobis $(s, a)$ |
|---|---|---|---|---|
| HalfCheetah-m.e. | RADAC | $1.22 \pm 0.21$ | $0.37 \pm 0.16$ | $2.08 \pm 0.62$ |
| HalfCheetah-m.e. | ORAAC | $29.27 \pm 12.76$ | $13.33 \pm 20.37$ | $19.06 \pm 19.57$ |
| HalfCheetah-m.e. | ORAAC-Diff. | $35.44 \pm 25.93$ | $3.28 \pm 0.883$ | $30.32 \pm 23.35$ |
| Hopper-m.e. | RADAC | $1.21 \pm 0.21$ | $0.38 \pm 0.50$ | $5.58 \pm 0.26$ |
| Hopper-m.e. | ORAAC | $11.60 \pm 3.95$ | $36.66 \pm 1.47$ | $18.44 \pm 11.41$ |
| Hopper-m.e. | ORAAC-Diff. | $22.75 \pm 6.56$ | $39.64 \pm 12.43$ | $15.19 \pm 6.10$ |
| Walker2d-m.e. | RADAC | $0.96 \pm 0.45$ | $0.63 \pm 0.26$ | $3.12 \pm 0.80$ |
| Walker2d-m.e. | ORAAC | $5.57 \pm 3.31$ | $6.99 \pm 2.97$ | $7.76 \pm 1.92$ |
| Walker2d-m.e. | ORAAC-Diff. | $18.54 \pm 2.71$ | $16.22 \pm 3.86$ | $15.89 \pm 3.19$ |

mean normalized score versus $\text{CVaR}_\alpha$, with error bars denoting the standard error of the mean across seeds (Fig. 15).

On HALFCHEETAH-MEDIUM-REPLAY-V2, varying $\alpha$ across $\{0.05, 0.10, 0.20\}$ primarily affects the tail: the points move noticeably along the horizontal axis in $\text{CVaR}_\alpha$, while the mean normalized score stays in a very tight band, indicating that RADAC can reshape the lower tail of the return distribution with only a minor impact on average performance. In contrast, on WALKER2D-MEDIUM-REPLAY-V2 the frontier moves up and to the right as we adjust $\alpha$: the more tail-sensitive settings simultaneously improve $\text{CVaR}_\alpha$ and the mean normalized score, suggesting that on this task catastrophic low-return trajectories are frequent enough that suppressing them not only reduces tail risk but also raises average returns. Overall, these frontiers confirm that the BC+CVaR objective in RADAC exposes a smooth knob to trade off tail-risk and mean return, and that in some regimes increasing tail sensitivity can strictly improve both safety and performance.

# F. Experimental Details

## F.1. 2D Synthetic Task Details

**Risky–Bandit dataset** We generate $N = 10^4$ state–action–reward tuples with dummy zero states. Actions come from two modes: (i) Ring (80%): radius $0.9 \pm 0.04$; base reward $\mathcal{N}(9, 0.3^2)$; with probability 0.05 a trap penalty $-40$ is applied (heavy lower tail). (ii) Center (20%): $\mathcal{N}(\mathbf{0}, 0.1^2\mathbf{I})$; reward $\mathcal{N}(5, 0.3^2)$. Actions are clipped to $[-1, 1]^2$.

All methods train on the same static dataset; when a BC regularizer is required we use the standard loss of the underlying generator. RADAC adds the CVaR term from Eq. 11 to the diffusion/flow BC objective and backpropagates. For each trained policy we draw 1,000 action samples for visualization in Fig. 3.

## F.2. Stochastic-D4RL MuJoCo Suite

**Datasets** We adopt the *stochastic MuJoCo* protocol for risk-sensitive offline RL, following (Urpí et al., 2021). Policies are evaluated on

$$\{\text{HOPPER, WALKER2D, HALFCHEETAH}\} \times \{\text{MEDIUM-EXPERT, MEDIUM-REPLAY}\},$$

Compared to prior work, we prefer MEDIUM-EXPERT and MEDIUM-REPLAY to validate both *risk sensitivity* and *policy expressiveness* under multimodal action distributions. For training, we relabel per-transition rewards in the offline datasets to inject stochastic hazards (velocity or torso-pitch thresholds with Bernoulli penalties and early termination); *the same hazard model is used at evaluation.* All algorithms (CQL, CODAC, ORAAC, DiffusionQL/FQL, RADAC, etc.) are trained on these relabeled rewards; the hazard indicator is never provided as an input feature or mask, so no method receives privileged information about hazard locations. This ensures the critic and the policy are trained on the risk-aware rewards rather than only being tested under hazards.

**Settings** Each task defines a monitored signal and an additive Bernoulli penalty when a safety condition is violated; pose-based tasks also include an early-termination threshold.

- **HALFCHEETAH**: monitor forward velocity. Apply a penalty with probability $p = 0.05$ if the threshold is exceeded. Thresholds/penalties: MEDIUM-EXPERT/MEDIUM-REPLAY uses $v > 10.0$/ $v > 5.0$ with penalty $-70.0$. No early

termination. Max episode steps: 200.

- **HOPPER / WALKER2D**: monitor torso pitch angle. When $|\theta|$ leaves the healthy range, add a penalty with probability $p = 0.10$; terminate early if $|\theta| > 2|\tilde{\theta}|$. Max episode steps: 500.

  - **HOPPER**: healthy range $[-0.1, 0.1]$ rad; penalty $-50.0$ when $|\tilde{\theta}| > 0.1$; early termination if $|\theta| > 0.2$.
  - **WALKER2D**: healthy range $[-0.5, 0.5]$ rad; penalty $-30.0$ when $|\tilde{\theta}| > 0.5$; early termination if $|\theta| > 1.0$.

## F.3. Baselines: Implementation & Hyperparameters

We include six representative offline-RL baselines:

- **CQL** (Kumar et al., 2020) (value pessimism). Non-distributional conservative Q-learning baseline.

- **CODAC** (Ma et al., 2021) (distributional conservative learning). We primarily use the CVaR-optimizing specification ("CODAC-C", $\mathrm{CVaR}_{0.1}$ objective).

- **ORAAC** (Urpí et al., 2021) (offline risk-averse actor–critic). A distributional critic with imitation-regularized policy optimizing a coherent risk objective.

- **DiffusionQL** (Wang et al., 2023) (expressive risk-neutral diffusion policy).

- **FQL** (Park et al., 2025) (expressive risk-neutral flow-matching policy).

- **ORAAC-Diffusion (ORAAC-Diff.)** (Chen et al., 2025) (offline risk-averse prior-anchored perturbation with a diffusion behavior prior)

**Hyperparameter selection & tuning**    For each of baselines, we run all baselines ourselves and tune the following parameters or adopt authors' recommended settings, mirroring the practice in (Kumar et al., 2020; Ma et al., 2021; Park et al., 2025; Urpí et al., 2021; Wang et al., 2023).

- **FQL** (Park et al., 2025): we sweep the policy weight $\alpha \in \{1, 10, 30, 100, 1000\}$ per task and report the best-performing setting (selection by $\mathrm{CVaR}_{0.1}$ unless noted).

- **ORAAC-Diff.** (Chen et al., 2025): we tune the mixing coefficient $\lambda \in \{0.1, 0.2, 0.4, 0.6\}$ per task and report the best-performing setting, following the role of $\lambda$ as the main environment-dependent trade-off coefficient in the original method.

- **DiffusionQL** (Wang et al., 2023): we consider $\eta \in \{0.1, 0.5, 1.0\}$ for the BC coefficient. We use authors' recommended configuration for other parameters without retuning. We also used the best checkpoint of their model on each benchmark by following their protocol.

- **ORAAC** (Urpí et al., 2021): use the paper's recommended configuration, including its distributional critic, risk level $\alpha = 0.1$, and anchor/prior regularization setting.

- **CODAC** (Ma et al., 2021): use the paper's tuned settings for D4RL (risk level $\alpha = 0.1$) without further tuning.

- **CQL** (Kumar et al., 2020): use the standard conservative coefficient and implementation defaults for MuJoCo locomotion.

## F.4. Estimating OOD Action Rates and Detectors

At evaluation time we measure the fraction of evaluation *state–action* pairs produced by a policy that fall outside the empirical support of the offline dataset. Let $\mathcal{D} = \{(s_i, a_i)\}_{i=1}^N$ denote the offline dataset for a given task, and let $\{(s_t^{(\mathrm{eval})}, a_t^{(\mathrm{eval})})\}_{t=1}^T$ be all state–action pairs emitted across evaluation rollouts.

Actions are scaled to $[-1, 1]$ per dimension in MuJoCo, so distances in action space are computed directly in $\ell_2$. For detectors operating on joint $(s, a)$ features, we standardize each *state* dimension using dataset statistics (z-scoring), while actions remain in their native $[-1, 1]$ scale.

Given a detector that assigns an OOD indicator

$$\mathbf{1}_{\mathrm{OOD}}\big(s_t^{(\mathrm{eval})}, a_t^{(\mathrm{eval})}\big) \in \{0, 1\},$$

we define the OOD rate

$$\varepsilon_{\mathrm{act}} \;=\; \frac{1}{T} \sum_{t=1}^{T} \mathbf{1}_{\mathrm{OOD}}\big(s_t^{(\mathrm{eval})}, a_t^{(\mathrm{eval})}\big),$$

and report the mean and standard deviation over seeds (mean $\pm$ std), matching Table 2 and Table 6.

**State-conditioned 1-NN detector.** We use a local notion of action support conditioned on the evaluation state. For an evaluation state $s$, define the $k$-NN state neighborhood in the dataset $\mathcal{N}_k(s) \subset \{s_i\}_{i=1}^{N}$ (we use $k=10$), and the corresponding local action set

$$\mathcal{A}_{\mathcal{D}}(s) \;=\; \{a_i \mid s_i \in \mathcal{N}_k(s)\}.$$

For each evaluation pair $(s_t^{(\mathrm{eval})}, a_t^{(\mathrm{eval})})$, we compute the local 1-NN action distance

$$d_t^{(\mathrm{eval})} \;=\; \min_{a_i \in \mathcal{A}_{\mathcal{D}}(s_t^{(\mathrm{eval})})} \|a_t^{(\mathrm{eval})} - a_i\|_2.$$

We set the threshold $\tau = \kappa \cdot \mathrm{median}\{d_i\}$ with $\kappa=3$, where each dataset reference distance $d_i$ is computed in the same manner using a leave-one-out construction that excludes the query transition itself from the candidate neighbor set. The OOD indicator is

$$\mathbf{1}_{\mathrm{OOD}}\big(s_t^{(\mathrm{eval})}, a_t^{(\mathrm{eval})}\big) = \mathbb{I}\Big\{d_t^{(\mathrm{eval})} > \tau\Big\}.$$

This state-conditioned 1-NN rate is the quantity reported in Sec. 5.3. Nearest-neighbor queries are implemented via a KD–tree over states to retrieve $\mathcal{N}_k(s)$, followed by a local 1-NN query in action space over $\mathcal{A}_{\mathcal{D}}(s)$.

**Alternative detectors (joint $(s, a)$ space).** To check that the RADAC $<$ ORAAC trend is not an artifact of the 1-NN score, we also evaluate two additional detectors on the joint $(s, a)$ space. We concatenate standardized state features with raw actions and fit the detectors on $\{(s_i, a_i)\}_{i=1}^{N}$, then evaluate on $\{(s_t^{(\mathrm{eval})}, a_t^{(\mathrm{eval})})\}_{t=1}^{T}$:

- **Local Outlier Factor (LOF).** We use LOF with 20 neighbors and contamination level $0.01$ in the joint $(s, a)$ space, and flag an evaluation point as OOD when LOF predicts an outlier label.

- **Single-Gaussian Mahalanobis distance.** We fit a single multivariate Gaussian $\mathcal{N}(\mu, \Sigma)$ to the joint $(s, a)$ data (with a small diagonal jitter added to $\Sigma$ for numerical stability). For any point $x = [s; a]$ we compute the squared Mahalanobis distance

$$d_{\mathrm{Mah}}^2(x) \;=\; (x - \mu)^{\top} \Sigma^{-1}(x - \mu).$$

We set the OOD threshold $\tau$ as the upper $95\%$ quantile of $\{d_{\mathrm{Mah}}^2(x_i)\}_{i=1}^{N}$ estimated from up to $5 \times 10^4$ subsampled dataset points, and declare $(s_t^{(\mathrm{eval})}, a_t^{(\mathrm{eval})})$ OOD if $d_{\mathrm{Mah}}^2(x_t^{(\mathrm{eval})}) > \tau$.

*Table 7.* **RAMAC: hyperparameters.** We keep only the knobs that materially affect performance and stability. Values are our defaults; brackets show typical sweep ranges.

| Global | |
| --- | --- |
| Discount $\gamma$ | 0.99 |
| Batch size $B$ | 256 |
| Target update $\tau_{\text{target}}$ | 0.005 |
| Risk level $\alpha$ | 0.1 |
| **Critic (Deterministic IQN)** | |
| #Quantiles $N$ | 32 |
| Grid $\mathcal{T}_N$ | $\{(i - \frac{1}{2})/N\}_{i=1}^{N}$ (fixed) |
| Embedding dim | 128 |
| Critic LR | $3 \times 10^{-4}$ |
| Huber $\kappa$ | 1 (fixed) |
| Double IQN | enabled |
| **Actor (shared)** | |
| Actor LR | $3 \times 10^{-4}$ |
| BC weight $\lambda_{\text{BC}}$ | 1.0 |
| Risk weight $\eta$ | **RADAC**: 0.05 [0.02–0.1], **RAFMAC**: 1000 [100–1000] |
| Double critic clipping | **RADAC**:$[-150, 150]$ or $[-300, 300]$, **RAFMAC**:$[-300, 300]$ |
| **RADAC-specific** | |
| Reverse diffusion steps $T$ | 5 (VP schedule) |
| **RAFMAC-specific** | |
| Flow steps $K$ | 10 (Euler, $\Delta t{=}1/K$) |

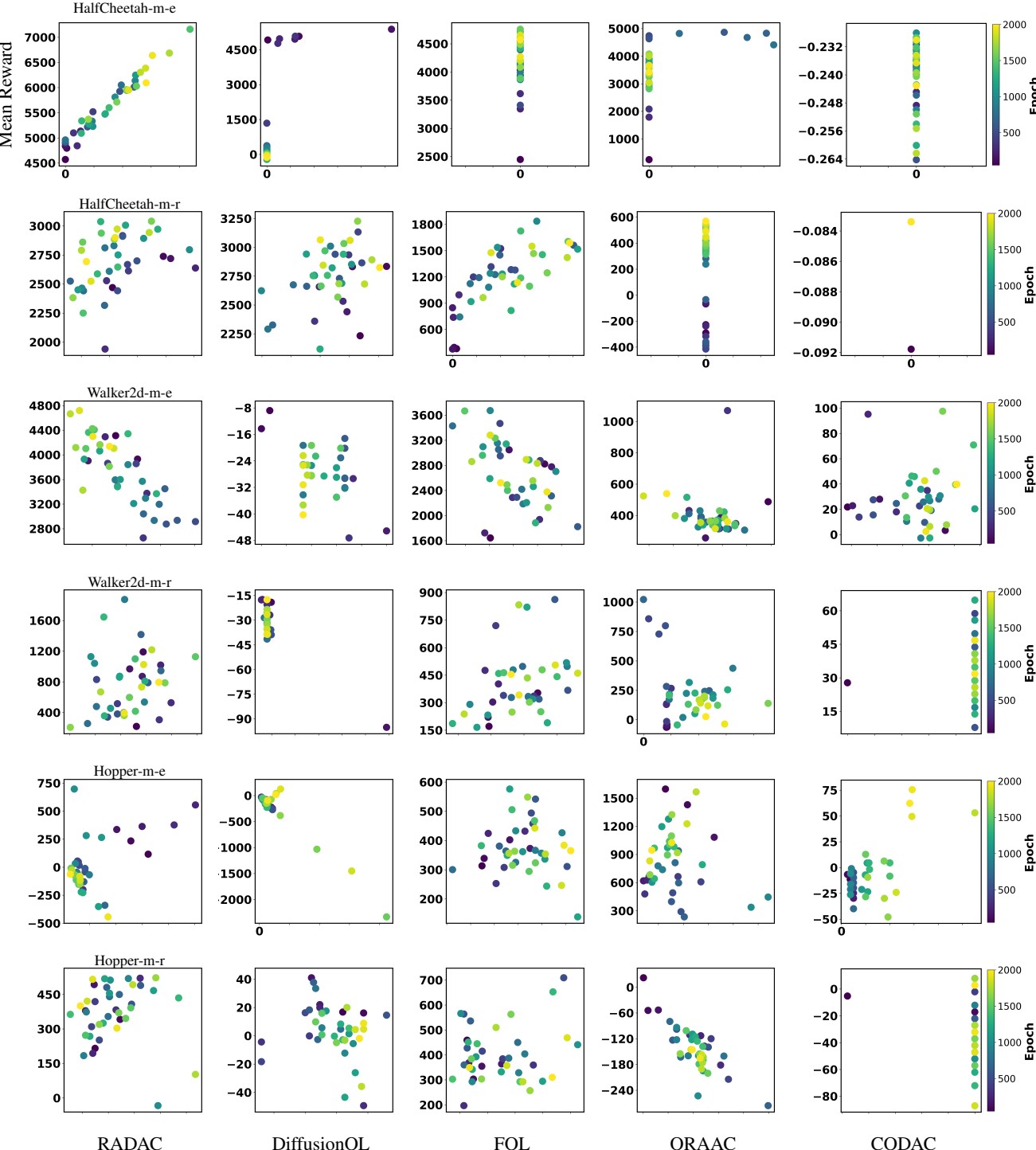

*Figure 10.* Pareto frontiers of return vs. safety violations. Rows are Stochastic–D4RL tasks (top→bottom: HALFCHEETAH-m-e, HALFCHEETAH-m-r, WALKER2D-m-e, WALKER2D-m-r, HOPPER-m-e, HOPPER-m-r); columns are algorithms (left→right: RADAC, DiffusionQL, FQL, ORAAC, CODAC). Points are evaluation snapshots across training (color encodes epoch; dark→yellow). $x$–axis: violation count per episode; $y$–axis: mean return (upper–left is better)

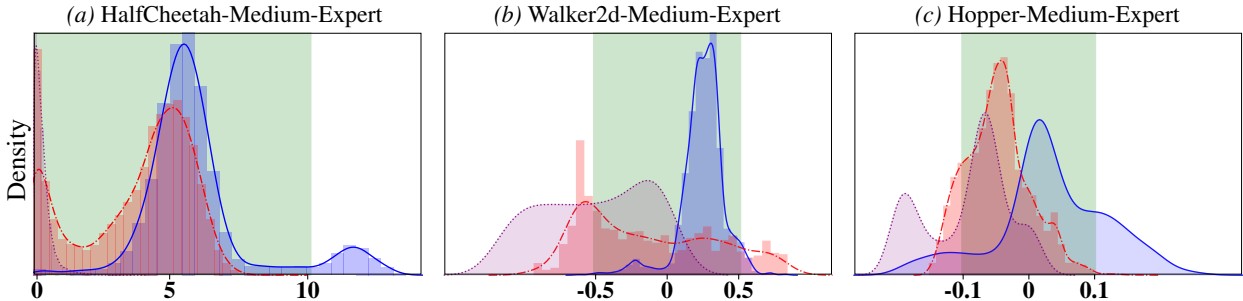

*Figure 11.* Additional qualitative safety heatmaps on MEDIUM–EXPERT. Shaded bands indicate risk-free operating ranges (HalfCheetah: $v \leq 10$; Walker2d: $|\theta| \leq 0.5$; Hopper: $|\theta| \leq 0.1$). The qualitative trends match Fig. 4 (main text): RADAC shifts mass away from hazardous regions while remaining concentrated on dataset-supported modes.

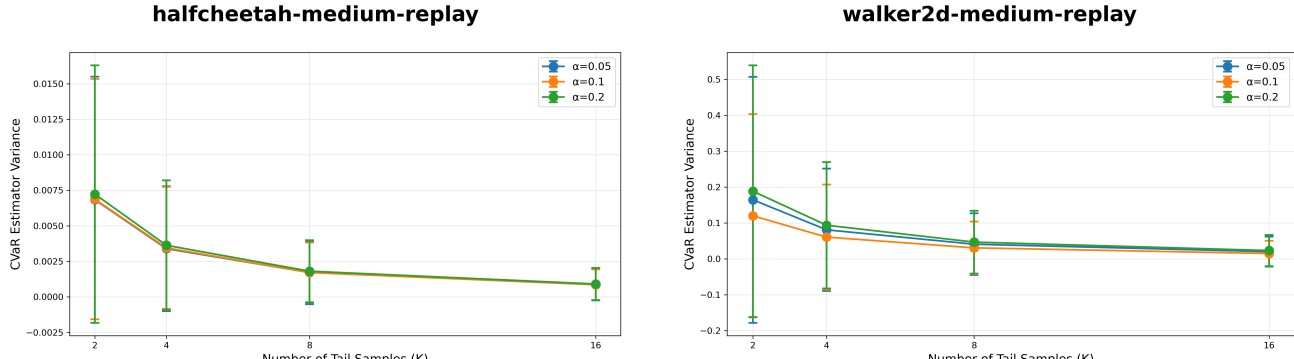

*Figure 12.* Empirical variance of the IQN-based $\mathrm{CVaR}_{0.1}$ estimator as a function of the tail sample size $K$ for the offline-selected RADAC policies on HALFCHEETAH-MEDIUM-REPLAY-V2 (left) and WALKER2D-MEDIUM-REPLAY-V2 (right). For each $K \in \{2, 4, 8, 16\}$ we compute $\widehat{\mathrm{CVaR}}_{\alpha}$; the curves show the mean per-state estimator variance with error bars indicating one standard deviation across states for $\alpha \in \{0.05, 0.1, 0.2\}$; all three risk levels exhibit a similar $1/K$-like decay.

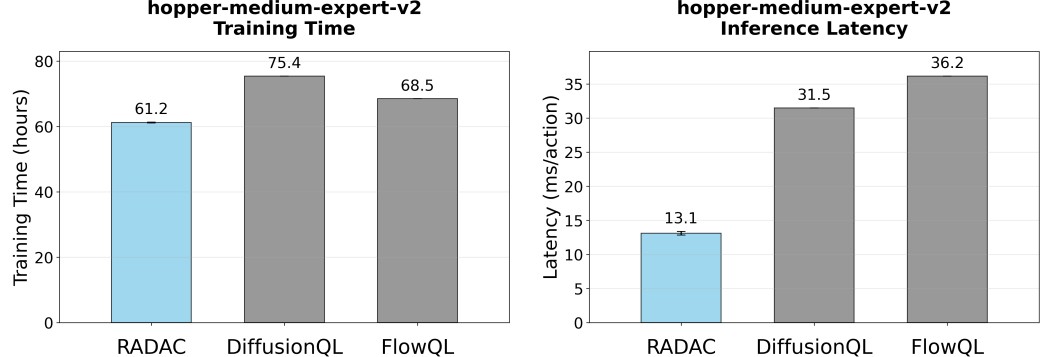

*Figure 13.* Wall-clock training time (left) and per-action inference latency (right) on HOPPER-MEDIUM-EXPERT-V2.

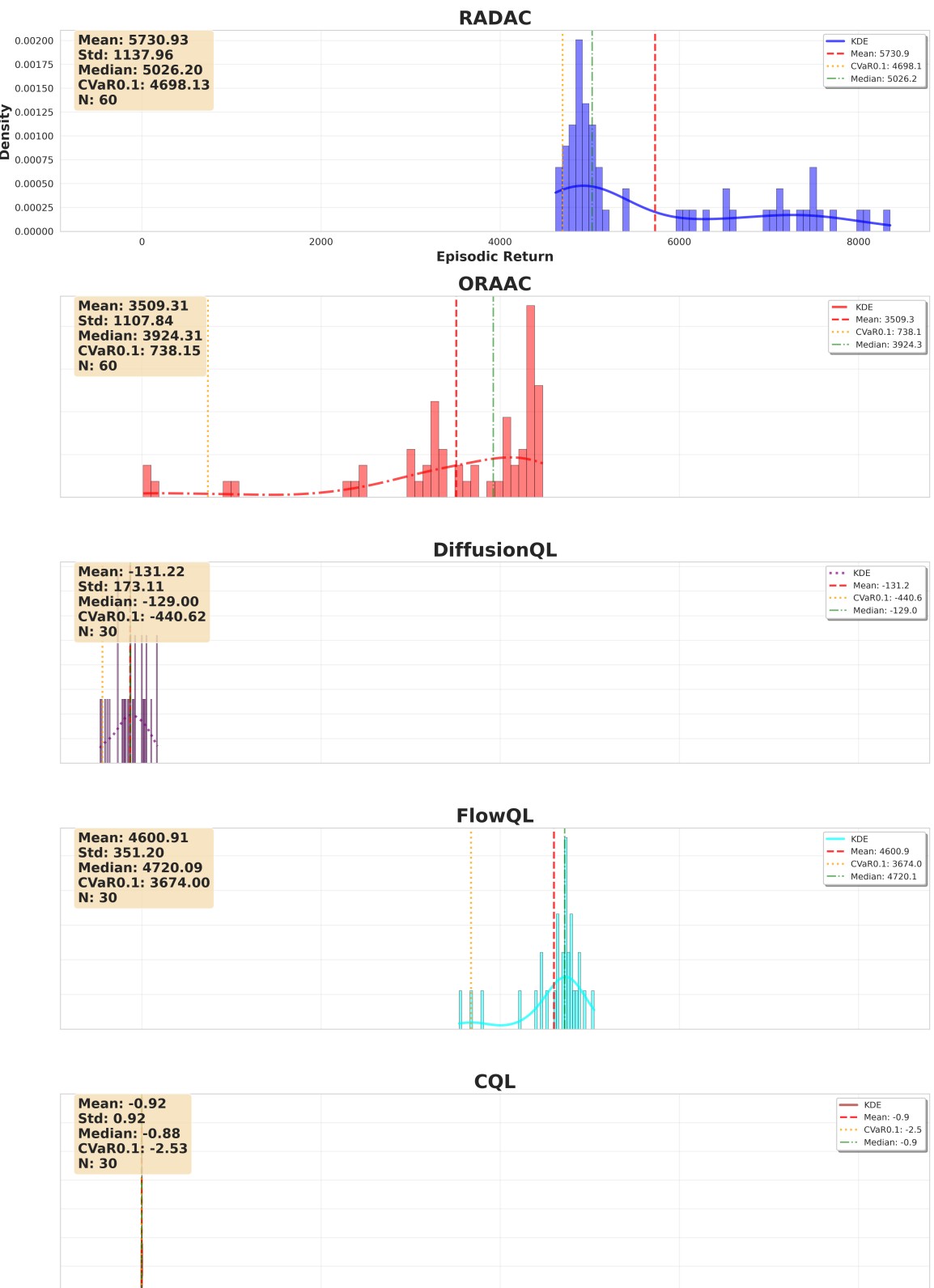

*Figure 14.* Empirical return distributions on HALFCHEETAH-MEDIUM-EXPERT-V2 under the stochastic hazard wrapper. Each subplot shows a histogram and KDE of episodic returns across evaluation rollouts, with vertical lines for the mean, median, and $CVaR_{0.1}$.

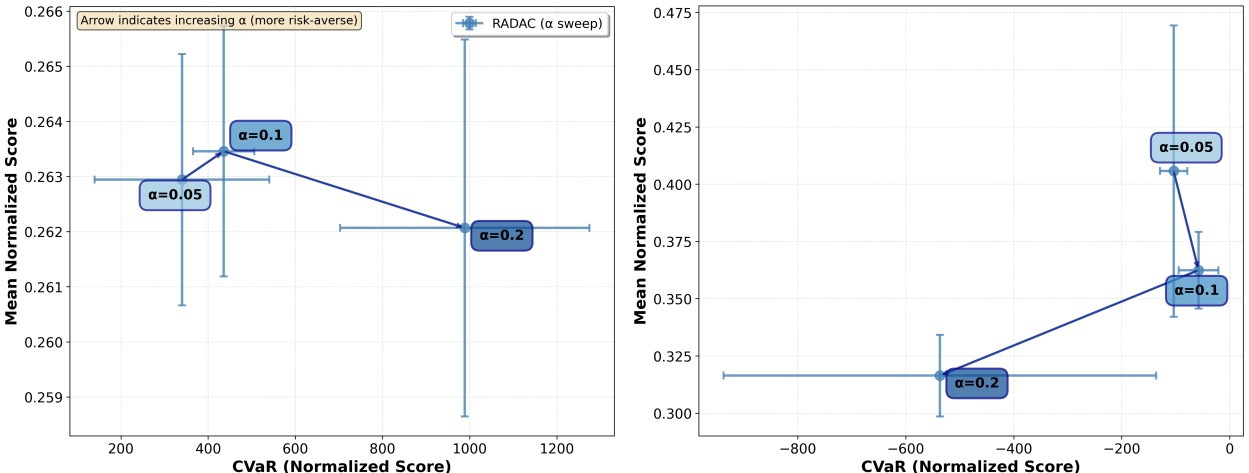

*Figure 15.* **Risk-return frontier for RADAC under different CVaR levels.** Each point shows the seed-averaged mean normalized score and $\text{CVaR}_\alpha$ for RADAC trained with a fixed $\alpha \in \{0.05, 0.10, 0.20\}$ on HALFCHEETAH-MEDIUM-REPLAY-V2 (left) and WALKER2D-MEDIUM-REPLAY-V2 (right). Error bars denote the standard error of the mean across seeds.

