# OpenReview forum: "RAMAC: Multimodal Risk-Aware Offline Reinforcement Learning and the Role of Behavior Regularization"
_ICML.cc/2026/Conference — ICML 2026 regular_

### Official Review · Reviewer_4JAQ · 2026-03-09

**Soundness:** 2
**Presentation:** 3
**Significance:** 3
**Originality:** 3
**Overall Recommendation:** 4
**Confidence:** 3

**Summary:**

This paper proposes RAMAC, a model-free offline RL framework that combines an expressive generative actor with a distributional critic and optimizes a single composite objective that mixes behavior cloning with a CVaR risk objective. The authors provide an objective-level analysis arguing that behavior regularization suppresses OOD actions and stabilizes CVaR under distribution shift, contrasting this with limitations of prior-anchored perturbation. Empirically, they instantiate RAMAC as RADAC and evaluate on a toy risky bandit and Stochastic-D4RL locomotion tasks, reporting improved while maintaining strong mean returns and lower measured OOD action rates.

**Compliance With Llm Reviewing Policy:**

Affirmed.

**Final Justification:**

This paper makes a technically solid contribution to risk-sensitive offline RL by combining an expressive generative actor with a distributional critic through a simple composite objective. I find the core idea original and timely, and the method appears broadly reusable across diffusion/flow-based actors. The paper is also generally clear, and the theoretical discussion helps motivate why behavior regularization can improve robustness under distribution shift. The empirical results on Stochastic-D4RL are meaningful and overall support the method’s effectiveness.

My main concerns in the original review were about the mechanism claim relating BC regularization, OOD control, and CVaR stability, the practical choice of the risk weight η, the fairness of the ORAAC comparison, and the interpretation of Lemma 4.1. The rebuttal addressed these concerns satisfactorily: the added ablations provide clearer main-task evidence for the proposed mechanism, the discussion of η gives useful practical guidance, the additional ORAAC sweeps improve confidence in baseline fairness, and the scope of Lemma 4.1 is now clarified more appropriately.

Overall, the rebuttal reinforced rather than changed my prior assessment. I therefore keep my score at Weak Accept: the paper is sound, reasonably original, and likely useful to researchers in this area, although its impact is still somewhat limited by the scope of evaluation.

**Key Questions For Authors:**

My questions are integrated into the "Weaknesses" section above.

I am willing to raise my score if the authors can satisfactorily address these concerns in the rebuttal.

**Limitations:**

Yes

**Strengths And Weaknesses:**

### Strengths:
- The paper precisely identifies one problem: the potential of expressive generative policies in risk-sensitive tasks has not been fully explored. Combining the two is a direction that is both innovative and aligns with current research trends. RAMAC appears broadly reusable across diffusion/flow actors.

- The paper not only presents the method but also provides solid theoretical analysis, explaining from a theoretical perspective why a simple behavioral cloning term can enhance safety.

- The experiments were conducted on the widely recognized risk-sensitive offline reinforcement learning benchmark, Stochastic-D4RL, covering multiple tasks. A comprehensive comparison was made against various baselines.

### Weaknesses:
- The optimal value of the risk weight \eta varies significantly across different environments. Does the paper offer any heuristic rules or adaptive methods to adjust \eta? How can the risk level be systematically chosen to achieve a desirable balance between average return and tail risk?

- In Lemma 4.1, do you believe the density lower bound assumption (p(a|s) \geqslant c > 0) on a ball is realistic for your implemented ORAAC(-Diffusion/-Flow)? If not, how should the lemma be interpreted in relation to Figure 3e–g?

- One of the paper’s arguments is that “BC regularization stabilizes the conditional value at risk by controlling OOD actions.” However, this mechanism is not demonstrated in the main tasks.

- ORAAC did not conduct experiments on the m-r and m-e. Is it reasonable to use its default parameter settings? Why did the original paper adjust parameters for FlowQL but not for ORAAC?

---

> ### Author Rebuttal · Authors · 2026-03-31
>
> Thank you for the thoughtful review and constructive feedback. We especially appreciate your questions on mechanism evidence, the choice of the risk weight $\eta$, and ORAAC fairness.
>
> ---
>
> * **The paper argues that BC regularization stabilizes CVaR by controlling OOD actions, but this mechanism is not demonstrated on the main tasks.**
>
> Thank you for highlighting this gap. Related to the BC-only / CVaR-only / full ablation in our response to Reviewer uEKX, we now make the main-task mechanism evidence explicit by re-evaluating OOD on the same ablations using the paper’s 1-NN metric.
>
> | Dataset         | Setting             | CVaR$_{0.1}$ | OOD (1-NN) |
> | --------------- | ------------------- | -----------: | ---------: |
> | Walker2d-m-r    | BC-only  |       -95.57 |       0.46 |
> | Walker2d-m-r    | RADAC (full)|       128.25 |       0.49 |
> | Walker2d-m-r    | CVaR-only               |       -53.04 |      62.87 |
> | HalfCheetah-m-r | BC-only |       269.54 |       2.42 |
> | HalfCheetah-m-r | RADAC (full)|       340.98 |       2.08 |
> | HalfCheetah-m-r | CVaR-only               |       -28.28 |      12.97 |
>
> These tasks show the intended two-stage pattern: **(i)** BC strongly suppresses OOD leakage, and **(ii)** within that low-OOD regime, the CVaR term improves the lower tail. Our claim is therefore not that OOD alone determines CVaR, but that BC suppresses support mismatch to make CVaR optimization stable and effective.
>
>
> * **About how  $\eta$ should be chosen**
>
>
> We do not propose an adaptive $\eta$ scheduler; instead, we view $\eta$ as a practical return-risk knob in the composite objective.
>
> This sensitivity is also mild in our submitted RADAC baselines: 5 of the 6 reported tasks use the same default $\eta=0.05$, and only HalfCheetah-m-r uses $\eta=0.02$.
>
> The paper already gives two forms of guidance: Fig. 9 shows the directional role of $\eta$ ($\eta\to 0$ becomes BC-like; larger $\eta$ emphasizes the RL/risk term), and Fig. 15 shows how different risk settings trace different return-risk operating points.
>
> To validate this interpretation on a main benchmark task (Walker2d-m-r, 3 seeds each), we additionally ran a same-task sanity check:
>
>
> | Setting | Mean return | CVaR$_{0.1}$ |
> | --- | ---: | ---: |
> | BC-only ($\eta=0$) | 617.31 | -95.57 |
> | RADAC ($\eta=0.02$) | 502.36 | 94.71 |
> | RADAC (default $\eta=0.05$) | 585.94 | 128.25 |
> | CVaR-only | -14.36 | -53.04 |
>
>
> These results support viewing $\eta$ as a practical operating knob rather than a brittle, heavily retuned hyperparameter: positive $\eta$ improves the lower tail, while different positive values move the policy across return-risk trade-offs. In practice, we recommend choosing the smallest $\eta$ that satisfies the target tail-risk tolerance; for tasks of similar scale with CVaR$_{0.1}$, $\eta=0.05$ is a reasonable default, with tuning around [${0.02, 0.05, 0.07, 0.1}$].
>
> * **Is the default parameter settings of ORAAC on m-e and m-r benchmarks reasonable?**
>
> Thank you for raising this fairness concern. For ORAAC, the key baseline-specific hyperparameter is the regularization weight $\lambda$, so to address the asymmetry concern we performed a bounded sanity sweep on one representative m-r task and one representative m-e task (3 seeds each, $\lambda\in\{0.1,0.25,0.5\}$):
>
> | Task | $\lambda$ | Mean return | CVaR$_{0.1}$ |
> | --- | ---: | ---: | ---: |
> | Walker2d-m-r | 0.10 | 300.10 | -133.63 |
> | Walker2d-m-r | **0.25** | 197.39 | -228.44 |
> | Walker2d-m-r | 0.50 | -19.45 | -71.78 |
> | Hopper-m-e| 0.10 | 455.74 | 153.13 |
> | Hopper-m-e | **0.25** | 708.24 | 378.31 |
> | Hopper-m-e | 0.50 | 314.50 | 168.84 |
>
> These results match the original ORAAC paper’s qualitative picture: $\lambda=0.25$ is not universally optimal, but it is also not arbitrary or pathological, and larger $\lambda$ can make the method overly pessimistic and hurt performance. Even the best case still trails RADAC. We therefore view the original ORAAC setting as a reasonable fixed-protocol baseline, supplemented with this sanity check.
>
> [1] Urpi et al., *Risk-Averse Offline Reinforcement Learning* (2021)
>
> * **About Lemma 4.1**
>
> We agree that the condition $p(a\mid s)\ge c>0$ on an entire ball is a **strong local assumption**, and we do **not** intend to claim that all implemented ORAAC-style actors literally satisfy such a density floor.
>
> Our intended interpretation is narrower: Lemma 4.1 is a **mechanism result**. It says that if an anchor-centered perturbation policy places nontrivial mass on a ball overlapping the complement of the behavior support, then its per-state OOD probability cannot be driven to zero merely by further residual training. Accordingly, Fig. 3e–g should be read as a theory-toy illustration of this geometric leakage mechanism, not as a literal verification of that density-floor assumption for each baseline. We will clarify this scope in the camera-ready revision.
>
>
> ---
>
> We hope these additions satisfactorily address your concerns and are helpful for your reassessment.

---

> > ### Author Rebuttal · Reviewer_4JAQ · 2026-04-02
> >
> > I appreciate the authors' response. My concerns are resolved, and I am keeping my score.

---

### Official Review · Reviewer_SghD · 2026-03-10

**Soundness:** 3
**Presentation:** 3
**Significance:** 3
**Originality:** 2
**Overall Recommendation:** 4
**Confidence:** 3

**Summary:**

The paper proposes a risk-aware policy learning framework for actor-crtici algorithms through intgegrating behavioral cloning and CVaR data.

**Compliance With Llm Reviewing Policy:**

Affirmed.

**Key Questions For Authors:**

1) What is the complexity of the proposed algorithm?
2) Can we compare this method with general methods in constrained RL?

**Limitations:**

1) The setting is restricted to CVaR risk and does not account other types of risk such as variance, barrier, etc.

**Strengths And Weaknesses:**

Strengths:
1) Empowers the well-established policy-based actor-critic algorithm to account for risk and provides theoretical analyses.
2) Tests on widely accepted domains and against several important baselines.

Weaknesses:
1) Restricts to the CVaR type of risk.

---

> ### Author Rebuttal · Authors · 2026-03-31
>
> Thank you for the thoughtful review. We appreciate your questions on scope, complexity, and comparison setting, as they help us sharpen what the paper is and is not claiming.
>
> ---
>
> * **The setting seems restricted to CVaR risk.**
>
> Thank you for raising this point. We would like to clarify that our paper is positioned in risk-sensitive offline RL, rather than as a general constrained-RL method. In this literature, **CVaR is a standard and representative objective**, and several closely related offline methods are likewise built around CVaR or distorted expectations, e.g., ORAAC, CODAC, 1R2R, and UDAC [1–4].
>
> Our contribution is therefore not that CVaR itself is new, but that we show how a single BC + risk objective can be used to train an expressive generative actor for offline risk-aware control, while avoiding the limitations of prior-anchored perturbation.
>
> Also, RAMAC is not conceptually tied only to CVaR. In App. E.3.3, we already report a **risk-distortion ablation** with CVaR, Wang, and CPW for RADAC on Table 4:
>
> | Distortion | HC-m-r Mean | HC-m-r CVaR$_{0.1}$ | W2d-m-r Mean | W2d-m-r CVaR$_{0.1}$ |
> | ---------- | ----------: | ------------------: | -----------: | -------------------: |
> | CVaR       |      2758.5 |          1759.5 |    681.3 |               -395.1 |
> | Wang       |      2653.5 |               310.8 |        417.3 |            -52.1 |
> | CPW        |  2777.9 |              1061.6 |         64.3 |               -203.6 |
>
> This is why the paper uses CVaR as the main instantiation: it is the most standard distortion in prior offline risk-aware RL, and our ablation shows that RAMAC is not restricted to CVaR alone. The three distortions trace different return-risk trade-offs: CVaR gives the strongest lower-tail improvement on HalfCheetah-Medium-Replay and the best mean on Walker2d-Medium-Replay, Wang tends to be more mean-oriented with weaker tail shaping, and CPW is intermediate but less stable across seeds. We provided the full details in App. E.3.3
>
> * **What is the complexity of the proposed algorithm?**
>
> RAMAC keeps the same overall structure as recent expressive actor-critic offline RL methods: each iteration performs one critic update and one actor update (Algorithm 1). Relative to risk-neutral expressive baselines, the main additional cost comes from (i) using a distributional critic instead of a scalar critic, and (ii) averaging lower-tail quantiles for the CVaR objective. It does not require an extra dynamics model, ensemble, or separate constrained optimization loop.
>
> Practically, the cost is still dominated by the expressive generative actor backbone (diffusion / flow), not by the CVaR aggregation itself. We already report **runtime and inference-latency comparisons** in Fig. 13 in App.E.4, which show that the distributional-critic + CVaR extension does not introduce an order-of-magnitude overhead.
>
> * **Can we compare this method with general methods in constrained RL?**
>
> We agree this is an interesting related direction. However, we believe it is not the primary apples-to-apples comparison for the problem formulation studied here. General constrained RL typically optimizes return subject to explicit cost / feasibility constraints, whereas our paper studies distributional tail-risk control in offline RL through a BC-regularized distorted-return objective [5].
>
> For this reason, we compare primarily against representative **risk-aware offline RL** baselines (e.g., O-RAAC, CODAC) and expressive offline RL baselines, which more directly match our setting. We agree that connecting RAMAC to explicit constrained-RL formulations would be a valuable future direction, especially when safety budgets or barrier-type constraints are available. We will clarify this in the camera-ready version.
>
>
>  [1] Urpi et al., *Risk-Averse Offline Reinforcement Learning* (2021).
>
>  [2] Ma et al., *Conservative Offline Distributional Reinforcement Learning* (2021).
>
> [3] Rigter et al., *One Risk to Rule Them All: A Risk-Sensitive Perspective on Model-Based Offline Reinforcement Learning* (2023).
>
> [4] Chen et al., *Uncertainty-aware Distributional Offline Reinforcement Learning* (2024).
>
> [5] Wachi et al., *A Survey of Constraint Formulations in Safe Reinforcement Learning* (2024).
>
>
> ---
>
> We hope this clarifies the intended scope and contribution of the paper. Please feel free to let us know if you have any additional questions or concerns. If we have addressed your concerns, would you consider raising your rating?

---

> > ### Author Rebuttal · Reviewer_SghD · 2026-03-31
> >
> > I have some concerns regarding the assumption that risk-aware RL is often tied to CVAR, and I shall cite some references below:
> >
> > - Variance Penalized On-Policy and Off-Policy Actor-Critic
> >
> > - Mean-Variance Policy Iteration for Risk-Averse Reinforcement Learning
> >
> > - Algorithmic aspects of mean–variance optimization in Markov decision processes
> >
> > - Risk-Averse Trust Region Optimization for Reward-Volatility Reduction
> >
> > - Learning the Variance of the Reward-To-Go
> >
> > - Risk-Sensitive Reinforcement Learning: Near-Optimal Risk-Sample Tradeoff in Regret **
> >
> > - Entropic Risk Measure in Policy Search **
> >
> > ** Entopic utility can be approximated with a mean and a variance term.

---

> > > ### Author Response · Authors · 2026-03-31
> > >
> > > ---
> > >
> > > Thank you for the helpful follow-up and for pointing out these broader risk-sensitive RL references.
> > >
> > > ---
> > >
> > > * **About our use of CVaR and the scope of the claim**
> > >
> > > Thank you for this clarification. You are right that, in the broader risk-sensitive RL literature, risk sensitivity is not limited to CVaR; mean-variance, entropic-risk, and related formulations are also important, including the works you cite.
> > >
> > > Our intended claim was narrower than that. The paper is scoped to **offline lower-tail / catastrophic-risk control** with expressive generative actors and a distributional critic, rather than to the entirety of risk-sensitive RL. In that narrower setting, we use CVaR as a **primary instantiation** because it directly targets lower-tail outcomes and aligns with several of the closest offline risk-aware baselines considered in the paper, such as ORAAC, CODAC, 1R2R, and UDAC [1–4].
> > >
> > > So our intended claim is **not** that “risk-aware RL = CVaR,” but rather that CVaR is a natural primary objective for the lower-tail offline risk-control setting studied in this paper.
> > >
> > > * **The method is not conceptually restricted to CVaR alone**
> > >
> > > We also want to clarify that RAMAC is **not conceptually tied only to CVaR**. In App. E.3.3, we already report a risk-distortion ablation with **CVaR, Wang, and CPW** for RADAC (Table 4). This shows that the framework extends beyond a single distortion, while different choices induce different return-risk trade-offs.
> > >
> > > This is why we view CVaR as the main instantiation in the paper: it is a standard lower-tail objective in closely related offline risk-aware work [1–4], and it provides a direct and interpretable target for catastrophic-tail mitigation. The broader point of the paper is the **BC-regularized risk-aware training of expressive generative actors**, not exclusivity to one distortion.
> > >
> > >
> > > * **What we will clarify in the revision**
> > >
> > > We agree that our previous wording could be read too broadly. To avoid overgeneralization, we will revise the wording in the final version to make this scope distinction explicit: namely, that CVaR is our primary instantiation for offline lower-tail risk control, rather than the only meaningful notion of risk in RL.
> > >
> > > We hope this clarifies the intended scope and contribution of the paper. Please feel free to let us know if you have any additional questions or concerns. If we have addressed your concern, would you consider revising your assessment?
> > >
> > > [1] Urpi et al., *Risk-Averse Offline Reinforcement Learning* (2021).
> > >
> > > [2] Ma et al., *Conservative Offline Distributional Reinforcement Learning* (2021).
> > >
> > > [3] Rigter et al., *One Risk to Rule Them Al: A Risk-Sensitive Perspective on Model-Based Offline Reinforcement Learning* (2023).
> > >
> > > [4] Chen et al., *Uncertainty-aware Distributional Offline Reinforcement Learning* (2024).
> > >
> > > [5] Wachi et al., *A Survey of Constraint Formulations in Safe Reinforcement Learning* (2024).

---

### Official Review · Reviewer_yhfP · 2026-03-12

**Soundness:** 3
**Presentation:** 3
**Significance:** 2
**Originality:** 2
**Overall Recommendation:** 5
**Confidence:** 3

**Summary:**

This work aims to address the limitations of prior risk-averse offline RL algorithms, which often rely on overly pessimistic or restricted policies. It proposes Risk-Aware Multimodal Actor-Critic (RAMAC), a model-free framework that pairs an expressive generative actor with a distributional critic. The framework optimizes a composite objective combining Conditional Value-at-Risk (CVaR) to mitigate catastrophic lower-tail risks with Behavioral Cloning (BC) to suppress out-of-distribution actions. Theoretical analysis and empirical evaluations on Stochastic-D4RL benchmarks demonstrate that RAMAC effectively stabilizes risk-sensitive learning in complex multimodal scenarios, achieving consistent gains in CVaR metrics while maintaining strong average returns.

**Compliance With Llm Reviewing Policy:**

Affirmed.

**Final Justification:**

Given that I did not have really any major concerns and after reading the other reviewers' comment, I have decided to keep my score.

**Key Questions For Authors:**

Minor comments:
- The description for Figure 1 does not sufficiently guide the reader on what the "correct" or "ideal" behavior for the policy density should look like. Additionally, both axes in Figure 1 lack labels.

- In Eq. 5, I think it should be $da_t$.

**Limitations:**

Yes

**Strengths And Weaknesses:**

## Strengths:
- The paper addresses a critical gap in offline RL by combining highly expressive generative models with risk-averse objectives. This allows the model to capture complex behaviors while maintaining safety, a combination where prior methods have struggled.

- The authors provide a rigorous analysis explaining the limitations of prior-anchored perturbation methods and the advantages of their BC regularization approach.

- On the Stochastic-D4RL benchmarks, the diffusion-based instantiation (RADAC) consistently outperforms representative offline RL baselines in terms of CVaR while maintaining competitive mean returns.

## Weaknesses:
- I did not encounter any major weaknesses in the work. It is well-motivated and clearly written. One minor point is the sensitivity of the hyperparameter $\eta$ used to balance the two losses; however, this is a standard challenge for most methods in this domain.

---

> ### Author Rebuttal · Authors · 2026-03-31
>
> ---
>
> Thank you very much for the positive and thoughtful review. We are especially grateful for your encouraging assessment of the motivation, analysis, and empirical results.
>
> ---
>
> * **Comments on Figure 1 and Eq. 5**
>
> Thank you for pointing these out. We agree that Fig. 1 can guide the reader more clearly. In the camera-ready version, we will add axis labels and revise the caption / surrounding text to better clarify the intended behavior of the policy density. We also agree that Eq. 5 contains a typo, and it should read $d\mathbf{a}_t$. We will correct this in the camera-ready version.
>
> * **Sensitivity of $\eta$**
>
> Thank you for raising this practical point. We agree that sensitivity to $\eta$ is important. To keep this response concise, we address this in more detail in our responses to Reviewers 4JAQ and uEKX, where we provide additional clarification and same-task sensitivity evidence. In brief, we view $\eta$ as a practical return-risk operating knob, and the added results suggest that positive $\eta$ values move the policy along a reasonably smooth return-risk trade-off rather than requiring brittle retuning.
>
> ---
>
> We would like to thank you again for the encouraging review and helpful suggestions. We believe these clarifications strengthen the paper, and we will incorporate them in the camera-ready version. Please let us know if you have any additional questions or concerns.

---

> > ### Author Rebuttal · Reviewer_yhfP · 2026-04-03
> >
> > I appreciate the authors' response for planning to revise figure 1 for the camera ready. As I had mentioned in my rebuttal, I did not encounter any major issues with paper. So I am keeping my score.

---

### Official Review · Reviewer_uEKX · 2026-03-12

**Soundness:** 2
**Presentation:** 3
**Significance:** 3
**Originality:** 2
**Overall Recommendation:** 4
**Confidence:** 3

**Summary:**

The paper proposes an offline, risk-aware RL framework with expressive generative actors capable of modeling multimodal action distributions. This is motivated by settings in which multiple distinct actions may be plausible for the same state, while catastrophic failures make tail-risk control especially important.

The proposed objective combines two components: (1) a behavior cloning (BC) term, which regularizes the policy toward the dataset distribution and aims to reduce out-of-distribution (OOD) actions, and (2) a tail-risk aversion term based on CVaR, computed from a distributional critic and estimated via Monte Carlo sampling.

Intuitively, the CVaR-based objective encourages the policy to avoid poor lower-tail outcomes, potentially at the cost of some mean return, while the BC term helps suppress OOD actions and stabilizes CVaR optimization.

**Compliance With Llm Reviewing Policy:**

Affirmed.

**Final Justification:**

I found the paper promising from the start: the method is simple and well motivated, Figure 3 is effective, and the experiments show meaningful gains in the risk-aware setting, especially in lower-tail performance. My main concerns were about attribution and novelty: in particular, the missing ORAAC-Diffusion comparison on the main benchmark and the lack of a clean ablation separating the effects of the BC and CVaR terms made it hard to judge whether the improvements came from the proposed objective itself or from using a diffusion-based actor more generally.

The rebuttal substantially addressed these concerns. The added ORAAC-Diffusion results on the main Mean/CVaR benchmark provide the comparison I felt was missing, and the new ablations give a much clearer picture of the roles of the BC and CVaR components. These additions make the paper’s core contribution significantly better supported and sharpen the novelty claim from the combination rather than the individual ingredients. While I still think the contribution is somewhat incremental rather than fundamentally new, the revised evidence is strong enough that I updated my assessment from weak reject to weak accept.

**Key Questions For Authors:**

Why is ORAAC-Diffusion not included in Table 1? It seems to be one of the closest baselines and would help isolate the benefit of the proposed objective.

**Limitations:**

yes

**Strengths And Weaknesses:**

### Strengths
* The method is simple and well motivated. Figure 3 provides a clear and convincing illustration.
* The experiments show strong improvements in the risk-aware setting, especially in lower-tail performance.

### Weaknesses
* The novelty seems somewhat limited. Behavior cloning and CVaR-based objectives from a distributional critic are not new individually; the main contribution is their combination within an expressive generative actor framework for risk-aware offline RL.
* The paper lacks a clean ablation on the main benchmarks isolating the effects of the BC term and the CVaR term.
* As far as I understand, ORAAC-Diffusion is the closest baseline, yet it is only used for OOD analysis and not reported on the main Mean/CVaR benchmark. This is an important missing comparison, since it would better isolate the contribution of the proposed objective from the benefit of simply using a diffusion actor.

---

> ### Author Rebuttal · Authors · 2026-03-31
>
> Thank you for the thoughtful review and constructive feedback. We especially appreciate your highlighting the missing ORAAC-Diffusion comparison and the need for a cleaner ablation, as these sharpened the paper’s core attribution test.
>
> ---
>
> * **Closest diffusion-based comparator (ORAAC-Diffusion) is missing from the main Mean / CVaR$_{0.1}$ benchmark.**
>
> Thank you for this important point. We agree that ORAAC-Diffusion (UDAC) is one of the closest baselines for isolating the contribution of our objective from the benefit of simply using a diffusion actor. To address this directly, we additionally evaluated ORAAC-Diffusion in the same Mean / CVaR$_{0.1}$ format (5 seeds each) as Table 1.
>
> For fairness, we tuned the ORAAC-Diffusion mixing coefficient over $\lambda \in \{0.1, 0.2, 0.4, 0.6\}$ and report the best $\lambda$ for each task. As noted in the ORAAC-Diffusion paper, $\lambda$ is the key trade-off coefficient and is environment-dependent [1].
>
> With this closest diffusion-based comparator included, RADAC remains stronger on all six tasks in both Mean and CVaR$_{0.1}$. This directly addresses the attribution question in your review: the gain is not explained by simply using a diffusion actor, but by the RAMAC composite objective that combines BC anchoring with tail-risk optimization in the deployed expressive policy.
>
> | Task                      | ORAAC-Diffusion Mean | ORAAC-Diffusion CVaR$_{0.1}$ |  RADAC Mean | RADAC CVaR$_{0.1}$ |
> | ------------------------- | -------------------: | ---------------------------: | ----------: | -----------------: |
> | HalfCheetah-Medium-Expert |               650.70 |                       455.40 |  **916.64** |         **805.25** |
> | Walker2d-Medium-Expert    |               823.20 |                       317.92 | **1708.68** |         **573.22** |
> | Hopper-Medium-Expert      |               577.73 |                       240.48 | **1277.74** |         **800.64** |
> | HalfCheetah-Medium-Replay |               220.00 |                        44.93 |  **525.84** |         **278.65** |
> | Walker2d-Medium-Replay    |                 4.62 |                      -113.20 |  **615.94** |         **145.21** |
> | Hopper-Medium-Replay      |                25.68 |                      -131.91 |  **385.58** |          **-8.16** |
>
> [1] Chen et al., *Uncertainty-aware Distributional Offline Reinforcement Learning* (2024)
>
> We agree that this comparison should have been included in the main benchmark table, and we will incorporate it in the revised camera-ready version.
>
> * **The paper lacks a clean ablation on the main benchmarks isolating the effects of the BC term and the CVaR term.**
>
> Thank you for this suggestion. To address it directly, we added representative 3-way ablations on two main benchmark tasks, comparing the full objective against BC-only and CVaR-only variants across 3 seeds:
>
> | Dataset                   | Variant     |  Mean | CVaR$_{0.1}$ |
> | ------------------------- | ----------- | ----: | -----------: |
> | HalfCheetah-Medium-Replay | RADAC (full)       | 518.7 |        341.0 |
> |                           | BC-only     | 485.9 |        269.5 |
> |                           | CVaR-only   |   2.9 |        -28.3 |
> | Walker2d-Medium-Replay    | RADAC (full)        | 585.9 |        128.3 |
> |                           | BC-only     | 617.3 |        -95.6 |
> |                           | CVaR-only   | -14.4 |        -53.0 |
>
> These tasks show complementary aspects of the same mechanism. On HalfCheetah-Medium-Replay, the full objective outperforms both individual components. On Walker2d-Medium-Replay, BC-only attains reasonable mean return but leaves the lower tail severely degraded, while CVaR-only collapses. Together, these ablations support our interpretation of RAMAC: BC stabilizes on-support optimization, while the CVaR term materially improves lower-tail behavior. For OOD-based mechanism evidence on the main tasks, please also see our response to Reviewer 4JAQ.
>
> * **Novelty seems somewhat limited, since BC and CVaR-based objectives are not new individually.**
>
> We agree that BC and CVaR are not individually new. Our intended contribution is more specific: RAMAC shows that, for expressive generative actors in offline risk-aware RL, a single composite objective combining BC anchoring with tail-risk optimization works as a simple, modular alternative to prior-anchored perturbation. The added ORAAC-Diffusion comparison tests whether the gain comes from simply using a diffusion actor, and the clean ablations test whether it comes from either term alone. Together, these additions sharpen the contribution claim to the combination itself rather than to the individual ingredients.
>
> ---
>
> If the added closest-baseline comparison and ablation address your concerns, we would be grateful if you would consider revising your assessment.

---

> > ### Author Rebuttal · Reviewer_uEKX · 2026-04-02
> >
> > The rebuttal substantially addressed my main concerns by adding the missing ORAAC-Diffusion comparison on the main benchmark and a clearer ablation isolating the effects of the BC and CVaR terms. These additions make the paper's contribution much better supported and clarify that the gains are not merely from using a diffusion actor, but from the proposed composite objective. Accordingly, I increased my score.

---

### Decision · Program_Chairs · 2026-04-30

**Decision:**

Accept (regular)

**Comment:**

This paper proposes RAMAC, a risk-aware policy learning framework for actor–critic algorithms by integrating behavioral cloning and CVaR-based data. Overall, I find the contribution somewhat incremental, as it mainly combines existing techniques. However, both the reviewers and I agree that the empirical performance gains are compelling, suggesting that the work still constitutes a solid contribution to the machine learning community.